# Interpreting CLIP: Insights on the Robustness to ImageNet Distribution Shifts

**Jonathan Crabbé**                                    *jonathan.cr1302@gmail.com*
*University of Cambridge*
*(work done while at Apple)*

**Pau Rodríguez**                                      *pau.rodriguez@apple.com*
*Apple*

**Vaishaal Shankar**                                   *vaishaal_shankar@apple.com*
*Apple*

**Luca Zappella**                                      *lzappella@apple.com*
*Apple*

**Arno Blaas**                                         *ablaas@apple.com*
*Apple*

**Reviewed on OpenReview:** *https://openreview.net/forum?id=1SCptTFtmV*

## Abstract

What distinguishes robust models from non-robust ones? While for ImageNet distribution shifts it has been shown that such differences in robustness can be traced back predominantly to differences in training data, so far it is not known what that translates to in terms of what the model has learned. In this work, we bridge this gap by probing the representation spaces of 16 robust zero-shot CLIP vision encoders with various backbones (ResNets and ViTs) and pretraining sets (OpenAI, LAION-400M, LAION-2B, YFCC15M, CC12M and DataComp), and comparing them to the representation spaces of less robust models with identical backbones, but different (pre)training sets or objectives (CLIP pretraining on ImageNet-Captions, and supervised training or finetuning on ImageNet). Through this analysis, we generate three novel insights. Firstly, we detect the presence of outlier features in robust zero-shot CLIP vision encoders, which to the best of our knowledge is the first time these are observed in non-language and non-transformer models. Secondly, we find the existence of outlier features to be an indication of ImageNet shift robustness in models, since we only find them in robust models in our analysis. Lastly, we also investigate the number of unique encoded concepts in the representation space and find zero-shot CLIP models to encode a higher number of unique concepts in their representation space. However, we do not find this to be an indicator of ImageNet shift robustness and hypothesize that it is rather related to the language supervision.

# 1 Introduction

Large pretrained multimodal models, such as CLIP (Radford et al., 2021), have demonstrated unprecedented robustness to distribution shifts around ImageNet[1]. When used as zero-shot image classifiers, their performance on ImageNet (Deng et al., 2009) translates remarkably well to performances on natural shifts of ImageNet, such as, e.g., ImageNet-V2 (Recht et al., 2019). This led to many works analyzing *what actually causes this remarkable robustness of CLIP to shifts around ImageNet*, with Fang et al. (2022) establishing that the root cause of CLIP's robustness lies in the quality and diversity of data it was pretrained on. In particular, they find the robustness of CLIP to shifts around ImageNet to disappear when it is pretrained on ImageNet-Captions, a modification of ImageNet suitable for unsupervised language-vision pretraining.

While the cause of CLIP's robustness to ImageNet shifts is thus known, we set out to establish how exactly robustness manifests itself in features learned by the model. Finding feature patterns in the representation space that only appear in robust models is the first step towards a better understanding of the emergence of robustness. It is also key to diagnosing robustness in times when only limited knowledge about the (pre)training distribution or the shifts is available. To find robustness patterns in robust CLIP models, we leverage the various models provided by Ilharco et al. (2021) in the OpenCLIP repository, as well as non-robust CLIP models pretrained on ImageNet-Captions provided by Fang et al. (2022), and models we train in a supervised way on ImageNet from scratch. We analyze the *visual features* in these models by probing their last layer activation vectors with quantitative interpretability tools, such as kurtosis analysis of activation vectors (Elhage et al., 2023), singular value decomposition (SVD) of classifier weight matrices and concept probing of the representation space (Bau et al., 2017). Through this analysis, we distill insights on distinctive characteristics of CLIP model features and CLIP ImageNet distribution shift robustness.

**Our contributions.** **(1)** We show that many robust CLIP models have *outlier features*. These features stand out as their activation is typically several orders of magnitude above the average activation of features in the same layer. Interestingly, this observation also holds for robust CLIP models with ResNet backbones. To the best of our knowledge, this is the first time that outlier features are observed in non-language and non-transformer models (Dettmers et al., 2022; Elhage et al., 2023; Sun et al., 2024). We also show through an SVD analysis that outlier features are propagated to the logits of downstream classifiers, which results in what we call *privileged directions* that are crucial to model predictions. **(2)** Through a comparative analysis we find that outlier features are *robustness indicators* that distinguish robust models from their non-robust counterparts. Since none of the non-robust models exhibit them, we find outlier features to be an indicator whose presence can indicate that a model will be robust to distribution shifts around ImageNet. Privileged directions on the other hand, are not such an indicator, since they can also be found in some of the non-robust models. **(3)** We show that robust zero-shot CLIP models all encode a high number of *unique concepts* in their features. As a consequence, the features of robust models are highly polysemantic, which means that they superpose a large set of concepts. Surprisingly, we find through our comparative analysis that representations that are rich in concepts are not necessarily more robust, as this property can also be found in non-robust ImageNet-Captions pretrained CLIP models. This suggests that language supervision tends to enrich visual representations in human concepts.

# 2 Background on CLIP models and notation

In this section, we summarize the CLIP paradigm introduced by Radford et al. (2021) and introduce the notation used in later sections. We explain how CLIP models are trained and used for image classification.

**CLIP architecture.** A CLIP model consists in an image encoder $f_v : \mathbb{R}^{d_X} \to \mathbb{R}^{d_H}$ and a text encoder $f_t : \mathbb{R}^{d_T} \to \mathbb{R}^{d_H}$ mapping to the same representation space $\mathbb{R}^{d_H}$ of dimensions $d_H \in \mathbb{N}$. With our notations, $d_X = C \cdot H \cdot W$ for images with $C \in \mathbb{N}$ channels, height $H \in \mathbb{N}$ and width $W \in \mathbb{N}$. Similarly, $d_T = L \cdot V$ for texts with context length $L \in \mathbb{N}$ and vocabulary size $V \in \mathbb{N}$. Given a batch of (image, text) pairs

---

[1]Similar to Radford et al. (2021), we use CLIP as a name for the general training technique of unsupervised language-vision pretraining, not only for the specific models obtained by OpenAI.

$\mathbb{B} = \{(x_v^{(b)}, x_t^{(b)}) \in \mathbb{R}^{d_X} \times \mathbb{R}^{d_T} \mid b \in [B]\}$, where $B \in \mathbb{N}$ denotes the batch size and $[B] := \{n \in \mathbb{N} \mid 1 \leq n \leq B\}$, the encoders are trained to minimize the symmetric contrastive loss

$$\mathcal{L}(\mathbb{B}, f_v, f_t) := \frac{\mathcal{L}_v(\mathbb{B}, f_v, f_t) + \mathcal{L}_t(\mathbb{B}, f_v, f_t)}{2}$$

$$\mathcal{L}_v(\mathbb{B}, f_v, f_t) := -\sum_{b=1}^{B} \log \frac{\exp\left(\tau^{-1} \cos\left[f_v\left(x_v^{(b)}\right), f_t\left(x_t^{(b)}\right)\right]\right)}{\sum_{b'=1}^{B} \exp\left(\tau^{-1} \cos\left[f_v\left(x_v^{(b')}\right), f_t\left(x_t^{(b)}\right)\right]\right)}$$

$$\mathcal{L}_t(\mathbb{B}, f_v, f_t) := -\sum_{b=1}^{B} \log \frac{\exp\left(\tau^{-1} \cos\left[f_v\left(x_v^{(b)}\right), f_t\left(x_t^{(b)}\right)\right]\right)}{\sum_{b'=1}^{B} \exp\left(\tau^{-1} \cos\left[f_v\left(x_v^{(b)}\right), f_t\left(x_t^{(b')}\right)\right]\right)},$$

with cos denoting the cosine similarity. $\mathcal{L}_v$ and $\mathcal{L}_t$ are InfoNCE losses from Oord et al. (2018) with a learnable temperature parameter $\tau \in \mathbb{R}^+$. These losses induce the encoders to align the image and text embedding pairs in the representation space $\mathbb{R}^{d_H}$ through the nominator, while separating image and text embeddings of distinct samples through the denominator, which differs between $\mathcal{L}_v$ and $\mathcal{L}_t$.

**Building zero-shot classifiers.** Once the model is trained, we have access to an image and text encoder that tend to align images with their text description. This can be used to build a zero-shot image classifier that discriminates between a set of $K \in \mathbb{N}$ classes $k \in [K]$. The typical approach is to combine the name of the class $k$, together with a template, to create $x_t^{(k)}$. For instance, the class `lion` can be combined with the template `an image of a <class>` to yield the text description `an image of a lion`. To assign a class to an input image $x_v \in \mathbb{R}^{d_X}$, one can assign logits to each class $k \in [K]$ as follows:

$$\text{logit}^{(k)}(x_v) := \tau^{-1} \cos\left[f_v\left(x_v\right), f_t\left(x_t^{(k)}\right)\right]$$

$$= \left[W \frac{f_v(x_v)}{\|f_v(x_v)\|}\right]_k,$$

where $\|\cdot\|$ denotes the euclidean norm in $\mathbb{R}^{d_H}$ and the matrix $W \in \mathbb{R}^{K \times d_H}$ has elements

$$W_{kj} := \left[\tau^{-1} \frac{f_t\left(x_t^{(k)}\right)}{\|f_t\left(x_t^{(k)}\right)\|}\right]_j$$

for each $k \in [K]$ and $j \in [d_H]$. A zero-shot image classifier can thus be built from CLIP as the composition between the linear classification head $W$ and the CLIP image encoder (with normalization). For ease of notation, we use $f \equiv f_v$, $x \equiv x_v$, etc. in the remainder of this work, unless otherwise specified.

## 3 Robustness of CLIP models

In this work, we focus on the robustness of models to shifts from ImageNet (Deng et al., 2009) to the five natural distribution shifts considered by Fang et al. (2022), namely ImageNet-V2 (Recht et al., 2019), ImageNet-R (Hendrycks et al., 2021a), ImageNet-Sketch (Wang et al., 2019), ObjectNet (Barbu et al., 2019) and ImageNet-A (Hendrycks et al., 2021b).

**Measuring robustness.** There are different ways to measure robustness to a specific distribution shift from source distribution A to shifted distribution B. In some works, especially in the literature on invariant and robust learning, simply performance on both source and shifted distribution are reported (Arjovsky et al., 2019; Makar et al., 2022; Jiang & Veitch, 2022; Kumar et al., 2022), and their quotient can be straightforwardly computed to report robustness in a single metric (*What fraction of the original performance is maintained on the shifted data?*). On the other hand, in the literature investigating the robustness of CLIP to ImageNet shifts, typically a slightly more involved metric called *Effective Robustness* (ER) is reported (Taori et al., 2020; Fang et al., 2022). We choose to include both, and will find that similar insights about

the robustness of models can be derived from both of them when it comes to distribution shifts around ImageNet.

Lastly, we note that recently Shi et al. (2023) have investigated the robustness of models to the five natural distribution shifts of ImageNet, using additionally a different data distribution than ImageNet (YFCC-15M Thomee et al. (2016)) as the source distribution. In that case, none of the models investigated (including CLIP models) is significantly more robust than the other models. Therefore, we keep the focus of our analysis on the shifts from ImageNet to its five natural distribution shifts, where most zero-shot CLIP models are without doubt significantly more robust than their finetuned or supervised counterparts, as we will recap in the following.

**Model pool.** We run our analyses across five backbone architectures: ResNet50, ResNet101, ViT-B-16, ViT-B-32, ViT-L-14 (He et al., 2015; Dosovitskiy et al., 2020). For each architecture, the OpenCLIP repository (Ilharco et al., 2021) contains robust pretrained CLIP models on various pretraining datasets: the original (unreleased) OpenAI pretraining set (OpenAI, Radford et al. (2021)), YFCC-15M (Thomee et al., 2016; Radford et al., 2021), CC-12M (Changpinyo et al., 2021), LAION-400M, LAION-2B (Schuhmann et al., 2022), and DataComp (Cherti et al., 2023). Furthermore, Fang et al. (2022) provided us with checkpoints of the non-robust CLIP models pretrained on ImageNet-Captions for three different versions of ImageNet-Captions: a first version for which only the original titles from the internet associated to the images are used as text (ImageNet-Captions-t), a second version for which a concatenation of original titles and description was used (ImageNet-Captions-td), and a last version for which a concatenation of original titles, description, and tags was used (ImageNet-Captions-tdt, for more details see Fang et al. (2022)). We load the pretrained vision encoders of all available combinations of architecture and pretraining dataset, and construct a zero-shot classification model for ImageNet using the methodology described in Section 2. By finetuning each robust zero-shot model on ImageNet, we obtain classifiers with lower robustness than their robust zero-shot counterparts (Andreassen et al., 2021; Kumar et al., 2022; Wortsman et al., 2022a). Lastly, since the non-robust ImageNet-Captions models were only trained for a ResNet50 backbone, we obtain further non-robust models for the remaining architectures by training them as classification models on ImageNet from scratch, and obtain models with far lower robustness than the finetuned ones (Fang et al., 2022) (details on model finetuning and training can be found in Appendix H).

**Results.** For all models, we compute the test accuracy on ImageNet, on the five shifted datasets (averaged), their quotient, as well as the ER (for details on the computation of ER, see Appendix A). We report the two metrics of robustness (quotient of test accuracies $\%_{acc}$, and ER) in Figure 1a and Figure 1b (for the individual test accuracies that these metrics are calculated from, see Appendix B). We see that according to both metrics, for each type of backbone the zero-shot CLIP models from OpenCLIP have the highest robustness ('Robust zero-shot CLIP'), which decreases but remains significant when fine-tuned on ImageNet ('Fine-tuned CLIP'). On the other hand, ImageNet-Captions CLIP models ('Non-robust zero-shot CLIP') and ImageNet trained supervised models ('Supervised')[2] do not exhibit any ER and much lower robustness according to $\%_{acc}$, typically loosing more than half of their ImageNet accuracy on the shifted datasets ($\%_{acc}$ dropping below 50%).

> **Take-away 1.** In agreement with the literature, we find that for each architecture, zero-shot CLIP models that were pretrained on OpenAI, YFCC-15M, CC-12M, LAION-400M, LAION-2B, or Data-Comp, are the most *robust to ImageNet distribution shifts*. This robustness, while decreased, remains significant after fine-tuning on ImageNet for almost all pretraining datasets. On the other hand, zero-shot CLIP models pretrained on ImageNet-Captions, and supervised models trained on ImageNet, are *non-robust to ImageNet distribution shifts*.

---

[2]For the ViT-L-14, we were unable to train a supervised version to convergence from scratch on ImageNet (Dosovitskiy et al. (2020) and He et al. (2022) comment on the difficulties of training such an overparametrized model on ImageNet).

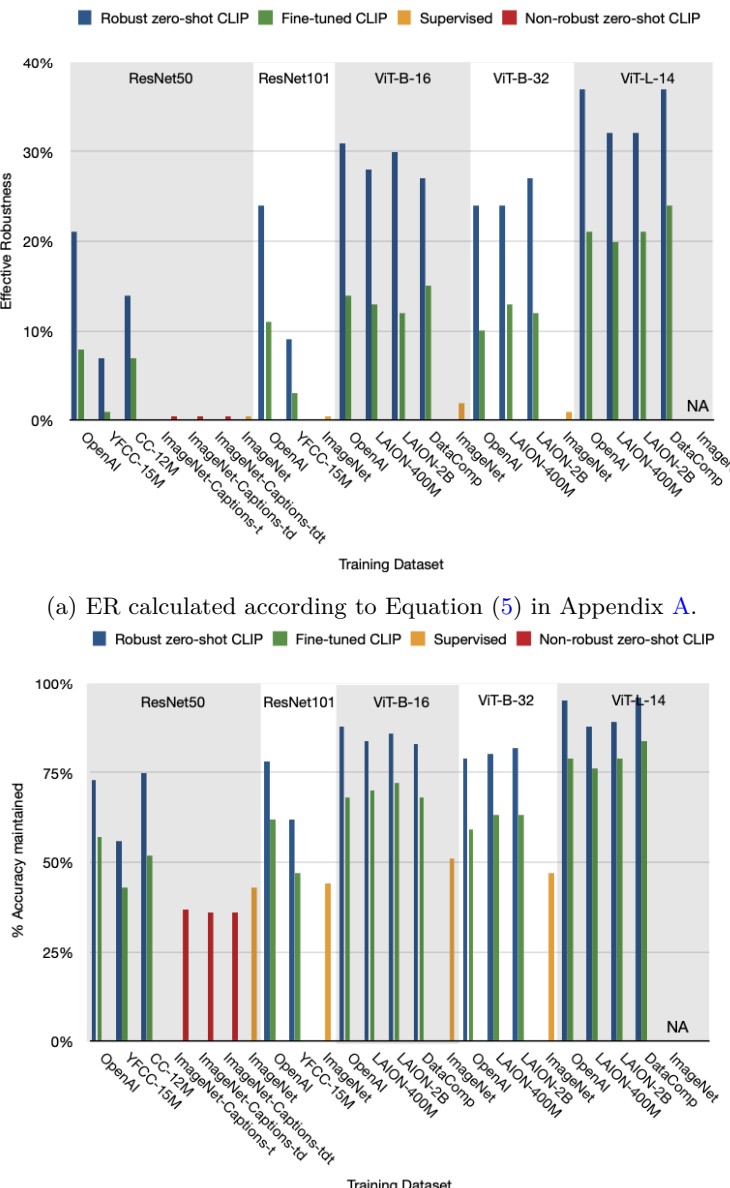

(a) ER calculated according to Equation (5) in Appendix A.

(b) $\%_{acc}$ calculated as the ratio between the average accuracy on the five ImageNet shifts and the ImageNet test accuracy.

Figure 1: *Robustness metrics for the models in our pool. Higher values indicate higher robustness. We see that for each backbone and pretraining data, robustness decreases from Robust zero-shot CLIP to Fine-tuned CLIP and reaches a minimum for ImageNet supervised ('Supervised') and Imagenet-Captions trained ('Non-robust zero-shot CLIP') models.*

## 4 Outlier features and privileged directions

In this section, we explain how we detect *outlier features* in zero-shot CLIP vision classifiers, and how we find them to be an indication of model robustness. We start by explaining what outlier features are and how they are surfaced, and then proceeed to analyse how they propagate to class logits in the form of privileged directions. We analyze both outlier features and privileged directions for each of the models in our model pool.

**Approach.**   We aim to analyze what models with high robustness have learned in comparison to models with lower robustness. To this end, we compare the features, i.e. the representation space spanned by the image encoder $f_v$ described in Section 2. Like Goh et al. (2021), we focus on the output of the encoder only since it is used for downstream classification by a linear head computing the ImageNet class logits. Furthermore, a *central kernel alignment* (CKA) analysis in Appendix C reveals that robust models differ from less robust models most consistently in this last layer, making it the most relevant layer to focus on from a comparative standpoint as well. We use the ImageNet test set to produce activation vectors $h^{(n)} = f_v(x^{(n)}) \in \mathbb{R}^{d_H}$ for each image $x^{(n)} \in \mathbb{R}^{d_X}$ fed to the encoder.

**Preliminary observations.**   Qualitatively observing the distribution of activation vectors, one thing that immediately stands out is the fact that some components $i \in [d_H]$ are much larger than the average activation: $h_i^{(n)} \gg d_H^{-1} \sum_{j=1}^{d_H} h_j^{(n)}$. A similar phenomenon has recently been observed by Dettmers et al. (2022) in large language models (LLMs). Such features, whose activation is substantially more important than average, were coined as *outlier features*. Subsequent work by Elhage et al. (2023) introduced a simple way to surface these outlier features, through a metric called *activation kurtosis*. We now use this criterion to quantitatively analyze the features of CLIP models.

**Activation kurtosis.**   Following Elhage et al. (2023), we measure the *activation kurtosis* to quantitatively evaluate the presence of outlier features in a model. The activation kurtosis is computed over all the components of an activation vector $h^{(n)}$, and averaged over $N$ activation vectors:

$$\text{Kurtosis} := \frac{1}{N} \sum_{n=1}^{N} \frac{1}{d_H} \sum_{i=1}^{d_H} \left[ \frac{h_i^{(n)} - \mu\left(h^{(n)}\right)}{\sigma\left(h^{(n)}\right)} \right]^4, \tag{1}$$

where $\mu(h) := d_H^{-1} \sum_{i=1}^{d_H} h_i$ and $\sigma^2(h) := d_H^{-1} \sum_{i=1}^{d_H} [h_i - \mu(h)]^2$. As explained by Elhage et al. (2023), Kurtosis $\gg 3$ indicates the presence of outlier features (3 being the kurtosis of an isotropic Gaussian).

We report the average kurtosis over the ImageNet test set in Figure 2a for each architecture and across the various levels of robustness. This leads to three insights. Firstly, across all architectures, there are robust zero-shot CLIP models with outlier features, as indicated by their Kurtosis $\gg 3$. Secondly, the kurtosis, like the robustness, drops when finetuning on ImageNet. In Appendix I we include an additional analysis showing that Kurtosis closely tracks the ER metric when interpolating between robust zero-shot CLIP models and fine-tuned CLIP models in weight space. Lastly and most importantly, the values Kurtosis $\approx 3$ obtained for the non-robust supervised, and non-robust zero-shot CLIP models reveal the absence of outlier features in non-robust models, suggesting that the presence of outlier features is an indicator of robustness that can only be found in robust models.

**Privileged directions in representation space.**   The strong presence of outlier features in the most robust models does not necessarily explain the performances of these models. Indeed, it is perfectly possible that outlier features are ignored by the linear head computing the class logits based on the activation vectors, e.g. if they are part of $\ker W$, the null space of the weight matrix $W$ defined in Section 2. Thus, to assess whether outlier features are of importance, we now introduce the notion of *privileged directions* of the representation space $\mathbb{R}^{d_H}$, as an instance of a generalized form of outlier features. While outlier features are studied in the canonical basis $\{e_1, \ldots, e_{d_H}\}$, since we can write $h_i = \text{Proj}_{e_i}(h)$, they can be generalized to be any set of directions of the representation space that receive a projection substantially above average (for more details and an illustration, see Appendix F).

We focus on the directions that are important for the computation of logits by the linear head $W$, namely *right singular vectors* of $W$. These can be identified by performing a singular value decomposition (SVD) of $W$, which can be written as $W = \sum_{i=1}^{\text{rank}(W)} \sigma_i \cdot u_i v_i^\intercal$, where $\sigma_i \in \mathbb{R}^+$, $u_i \in \mathbb{R}^{d_Y}$ and $v_i \in \mathbb{R}^{d_H}$ respectively correspond to *singular values* (SV), *left singular vectors* (LSV) and *right singular vectors* (RSV) of $W$. In this decomposition, each RSV $v_i$ corresponds to a direction in representation space that is mapped to the logits encoded in the LSV $v_i$. Since both of these vectors are normalized $\|u_i\| = \|v_i\| = 1$, the importance of the direction $v_i$ for $W$ is reflected by the SV $\sigma_i$. Note that the SV $\sigma_i$ by itself *does not* refer to the model's

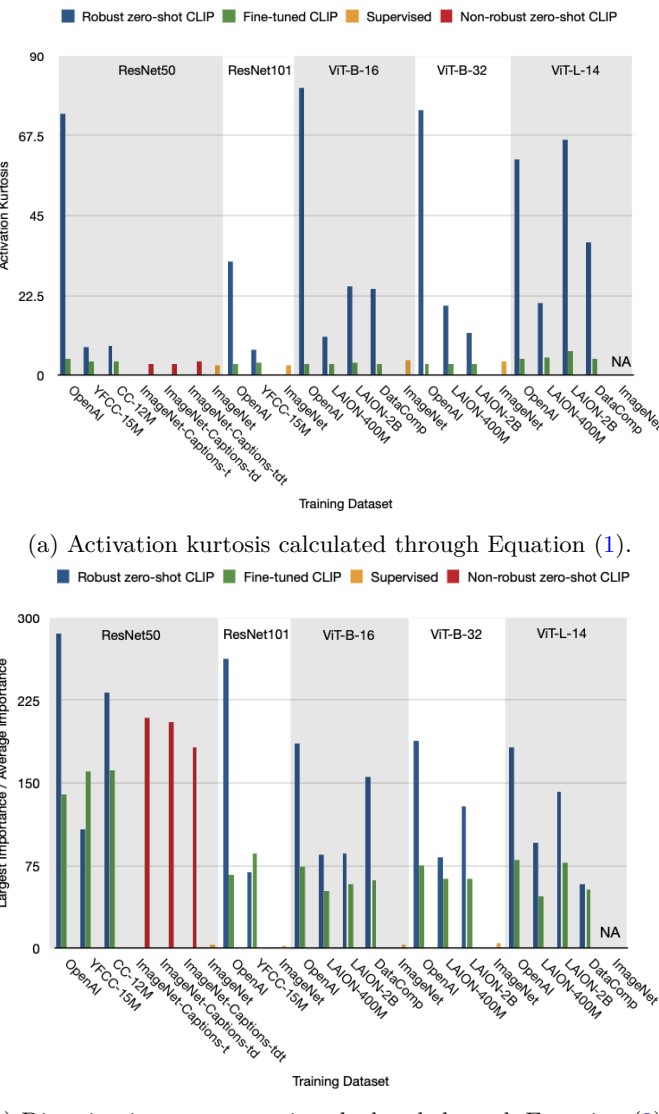

(a) Activation kurtosis calculated through Equation (1).

(b) Direction importance ratio calculated through Equation (3).

Figure 2: *Outlier features and privileged directions. Most robust zero-shot CLIP models have outlier features (high kurtosis) and privileged directions (high direction importance outlierness). Fine-tuned models have no outlier features but still exhibit privileged directions, although those are noticeably less privileged. Supervised models and non-robust zero-shot CLIP models have no outlier features. The full distribution of importance scores can be found in Appendix E.1*

encoder activations. How can we measure if the direction $v_i$ is typically given an important activation by the encoder? We propose to measure the average cosine similarity of activation vectors $h^{(n)}$ with this direction. This leads us to a unified importance metric for each direction $i \in [\text{rank}(W)]$ of the representation space:

$$\text{Importance}(i) := \underbrace{\frac{\sigma_i}{\sum_{j=1}^{\text{rank}(W)} \sigma_j}}_{\text{Classification head importance}} \cdot \underbrace{\sum_{n=1}^{N} \frac{|\cos(v_i, h^{(n)})|}{N}}_{\text{Encoder importance}} . \tag{2}$$

Note that this metric is defined so that $0 < \text{Importance}(i) < 1$. With this metric, we can measure to what extent the presence of outlier features induces privileged directions in representation space. If such privileged directions exist, we expect some singular directions $v_i$ to have an Importance($i$) substantially

higher than average, i.e. with $\text{Importance}(i) \gg \text{rank}(W)^{-1} \sum_{j=1}^{\text{rank}(W)} \text{Importance}(j)$. We thus can identify privileged directions as the RSVs associated to outlier values in the importance scores. Indeed, we observe the existence of such privilege directions in Figure 2b, where we plot the quotient of the largest and the average importance score for each model

$$\text{Importance Ratio} := \frac{\max_{i=1}^{\text{rank}(W)} \text{Importance}(i)}{\sum_{j=1}^{\text{rank}(W)} \text{Importance}(j)\text{rank}(W)}. \tag{3}$$

We notice that all the robust zero-shot CLIP models have such a privileged direction. For the less robust finetuned models, these privileged directions still exist, but not they are not as strong. This indicates that finetuning de-emphasizes privileged directions. Finally, the importance distributions of non-robust models have no privileged directions with only one exception: the ImageNet-Captions CLIP models. These models have privileged directions in the absence of outlier features, which might appear surprising at first. By analyzing Equation (2), we deduce that these models should have some singular values that are substantially larger than average. As explained in Section 2, these singular values correspond to a weight matrix obtained by stacking activations of the language encoder from the CLIP model. Since the non-robust ImageNet-Captions CLIP models show little to no robustness, we deduce that privileged directions can not serve as an indication of robustness.

> **Take-away 2.** Many robust models exhibit outlier features and privileged directions in their representation spaces. Since we do not find outlier features in non-robust models, outlier features appear to be an indication of model robustness to shifts around ImageNet. Privileged directions can however also be found in non-robust models and are thus not an indication of robustness.

**Emergence of outlier features.**   We have observed that outlier features are an indication of robustness to ImageNet distribution shifts. We would like to suggest a hypothesis to explain why they distinguish robust models from their less robust counterparts. In our analysis, they can only be found in robust zero-shot CLIP models that were typically trained on datasets with a size and diversity that is several orders of magnitude above that of ImageNet. Similarly, Sun et al. (2024) observed 'massive activations' (closely related to outlier features) only in transformers which were pretrained on datasets that were large and diverse in comparison to ImageNet, but not in transformers trained on ImageNet itself. Combined together, these two facts suggest that outlier features result from training on larger and more diverse datasets (in comparison to the evaluation dataset), and that robustness and outlier features thus share a common root cause in the type of pretraining data.

**Relationship of outlier features to pruning.**   Note that previous work on LLMs found that outlier features also have positive effects on model pruning: Sun et al. (2023) found that LLMs with outlier features can be efficiently pruned by retaining features with larger activations. We also found some weak evidence of this kind when pruning latent directions (see Appendix G).

## 5   Concept probing

In this section we find that robust CLIP models encode more concepts than non-robust supervised models. However, also non-robust CLIP models encode similarly high numbers of concepts. Our analysis thus suggests that this stems from language supervision in CLIP, rather than being related to robustness. We first describe our approach, and then discuss the concepts encoded in the privileged directions identified in the previous section. We then show that robust and non-robust CLIP models encode more unique concepts than fine-tuned and supervised models. Lastly, we explain how this leads to polysemanticity in CLIP models.

**Approach.**   With the discovery of privileged directions in the representation spaces of models with robustness to Imagenet shifts, it is legitimate to ask what type of information these directions encode. More generally, are there differences in the way robust models encode human concepts? To answer these questions, we use *concept probing*. This approach was introduced by Bau et al. (2017), along with the *Broden dataset*.

This dataset consists of $63,305$ images illustrating $C = 1,197$ concepts, including scenes (e.g. street), objects (e.g. flower), parts (e.g. headboard), textures (e.g. swirly), colors (e.g. pink) and materials (e.g. metal). Note that several concepts can be present in each image. For each concept $c \in [C]$, we construct a set of positive images $\mathcal{P}^c \subset \mathbb{R}^{d_X}$ (images that contain the concept) and negative images $\mathcal{N}^c \subset \mathbb{R}^{d_X}$ (images that do not contain the concept). In the following, we shall consider balanced concept sets: $|\mathcal{P}^c| = |\mathcal{N}^c|$. Concept probing consists in determining if activations in a given direction of the representation space discriminate between $\mathcal{P}^c$ and $\mathcal{N}^c$.

**Assigning concepts to directions.** We are interested in assigning concepts to each RSV $v_i$ of the linear head matrix $W$. To determine whether a representation space direction enables the identification of a concept $c \in [C]$, we proceed as follows. For each activation vector $h^{(c,n)} = f_v(x^{c,n})$ associated to positive images $x^{(c,n)} \in \mathcal{P}^c$, we compute the projection $\text{Proj}_{v_i}(h^{(c,n)})$ on the RSV. We perform the same computations for the projections $\text{Proj}_{v_i}(h^{(\neg c,n)})$ of negative images $x^{(\neg c,n)} \in \mathcal{N}^c$. If the direction $v_i$ discriminates between concept negatives and positives, we expect a separation between these projections: $\text{Proj}_{v_i}(h^{(c,n)}) \neq \text{Proj}_{v_i}(h^{(\neg c,n)})$. In other words, we expect the projections on $v_i$ to be a good classifier to predict the presence of $c$. Following Suau et al. (2022), we measure the average precision $\text{AP}_i^c$ of this classifier to determine whether the concept is encoded in direction $v_i$. We set a threshold $\text{AP}_i^c \geq 0.9$ to establish that the concept $c$ is encoded in $v_i$[3].

**Interpreting privileged directions.** We look at the concepts with highest $\text{AP}$ in the privileged direction of each robust model represented in Figure 2b (i.e. robust and fine-tuned CLIP). In the following, we illustrate our insights about which concepts are encoded in the privileged direction on the OpenAI pretrained models. Note that the following discussion generalizes well to the other robust models, for which we report the top-3 concepts in Appendix E.3.

Interestingly, the most privileged direction of both zero-shot OpenAI ViTs encode the same top-3 concepts: *meshed, flecked* and *perforated*. The most privileged direction of the ResNet50 also encodes concepts related to textures, with the *knitted* and *chequered* concepts. The most privileged direction of the ResNet101 encodes concepts with high colour contrasts, with the *moon bounce, inflatable bounce game* and *ball pit* concepts. We note that all these concepts describe regular alternating patterns, either through the presence/absence of holes or through the variation of colours.

Let us now discuss the concepts encoded in the privileged directions of finetuned models. We find that finetuning replaces the above texture-related concepts by less abstract and more concrete concepts. After finetuning, the concept that are best encoded in the privileged directions are *martial art gym* for the ViT-B/16, *tennis court* for the ViT-B/32, *mountain pass* for the ResNet50 and *flight of stairs* for the ResNet101. All of these concepts are substantially less generic than the ones encoded in the zero-shot models.

> **Take-away 3.** We qualitatively observe that privileged directions of robust zero-shot CLIP models tend to encode rather generic texture information, while fine-tuning tends to replace these generic concepts in privileged directions by less abstract and more concrete concepts.

**Number of unique concepts.** Let us now discuss the representation spaces of various models beyond privileged directions. A first way to characterize a representation space as a whole is to simply count the number of unique concepts they encode. In other words, for the representation space of each model, we evaluate

$$N_{\text{unique}} := |\mathcal{C}| \qquad \mathcal{C} := \{c \in [C] \mid \text{AP}_i^c \geq 0.9 \text{ for some } i \in [\text{rank}(W)]\}. \tag{4}$$

We report the number of unique concepts encoded in each type of model from our pool in Figure 3.

We notice that robust zero-shot CLIP models encode substantially more concepts than their fine-tuned and supervised counterparts, which thus at first might appear to be another indicator of robustness. However, also some of the non-robust zero-shot ImageNet-Captions pretrained CLIP models encode a relatively high number of unique concepts, namely ImageNet-Captions-t and ImageNet-Captions-td. On the other hand,

---

[3]All the below conclusions still hold in the same way if we change the threshold to other values such as 0.8, 0.85, or 0.95.

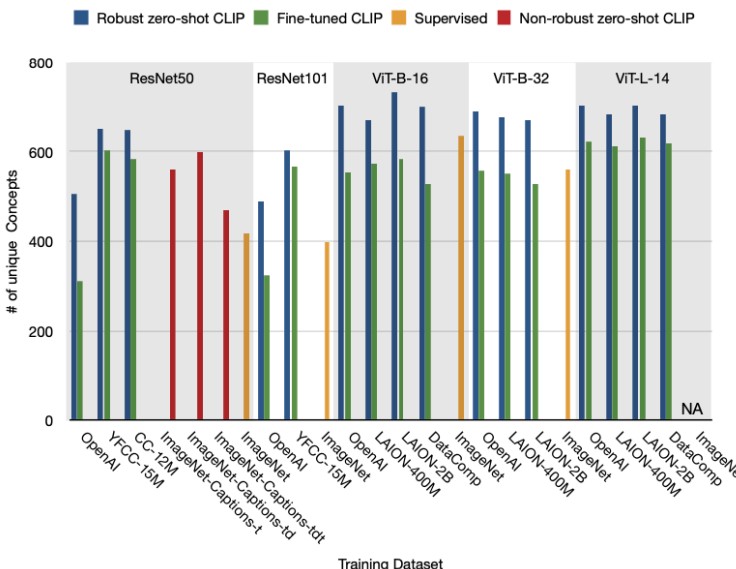

Figure 3: *Results of the unique concept analysis, showing total number of unique Broden concepts encoded in last layers, as in Equation* (4). *Zero-shot CLIP models encode substantially more concepts than supervised models.*

ImageNet-Captions-tdt, which also includes (potentially less diverse) image tags in the text associated to each image, does not encode that many concepts, in fact only barely more than the supervised ResNet50. Thus, the number of unique concepts in the model features seems to be more strongly influenced by the type and diversity of the text associated to images during training and their inclusion into the training objective, rather than by the model robustness or a root cause thereof.

We can further compare the set of concepts encoded for the other architectures, by producing their *Venn diagrams* in Figure 4 (ImageNet-Captions models omitted here for better comparability among backbones). In each case, the most significant section of the Venn diagrams is the overlap between all 3 model types (this ranges from 237 concepts for RN-101 to 516 concepts for ViT-B/16). This suggests that all 3 models share a large pool of features that are useful for each respective task the models were trained on. This is confirmed by Figure 5, where we track the size of overlap of the finetuned concepts with all sections of the Venn diagram (normalized by the number of finetuned concepts $|\mathcal{C}_{\text{fine}}|$) during finetuning. As we can see, finetuning increases the amount of concepts that are shared by the zero-shot, finetuned, and supervised models, since $\mathcal{C}_{\text{fine}} \cap \mathcal{C}_{\text{zero}} \cap \mathcal{C}_{\text{sup}}$ increases. Interestingly, the overlap with concepts specific to the supervised models $((\mathcal{C}_{\text{fine}} \cap \mathcal{C}_{\text{fine}}) \setminus \mathcal{C}_{\text{zero}})$ does not always increase substantially, which suggests that finetuning does not necessarily encode concepts that are exclusively learned in a supervised setting. In agreement with Figure 3, the Venn diagrams show that zero-shot models encode many concepts that are unknown to fine-tuned and supervised models (this ranges from 77 concepts for ViT-B/16 to 105 concepts for RN-50).

**Connection to polysemanticity.** A large number of encoded concepts can come at the cost of interpretability. As explained by Olah et al. (2020), superposing many concepts in a given representation space creates *polysemantic features*. Those features correspond to directions of the representation spaces that encode several unrelated concepts, which makes the interpretation of such features challenging. Polysemantic features are typically identified by using feature visualization to construct images that maximally activate the unit (neuron/representation space direction) of interest (Olah et al., 2017). A manual inspection of these images permits to identify that several concepts are present in the image maximizing the unit activation.

A manual feature visualizations for each RSV $v_i$ of each model in our pool would be prohibitively expensive. For this reason, we use a *proxy* for polysemanticity based on the Broden dataset. For each RSV $v_i$, we count the number of concepts encoded in the corresponding direction of the representation space $N_{\text{concept}}(i) :=$

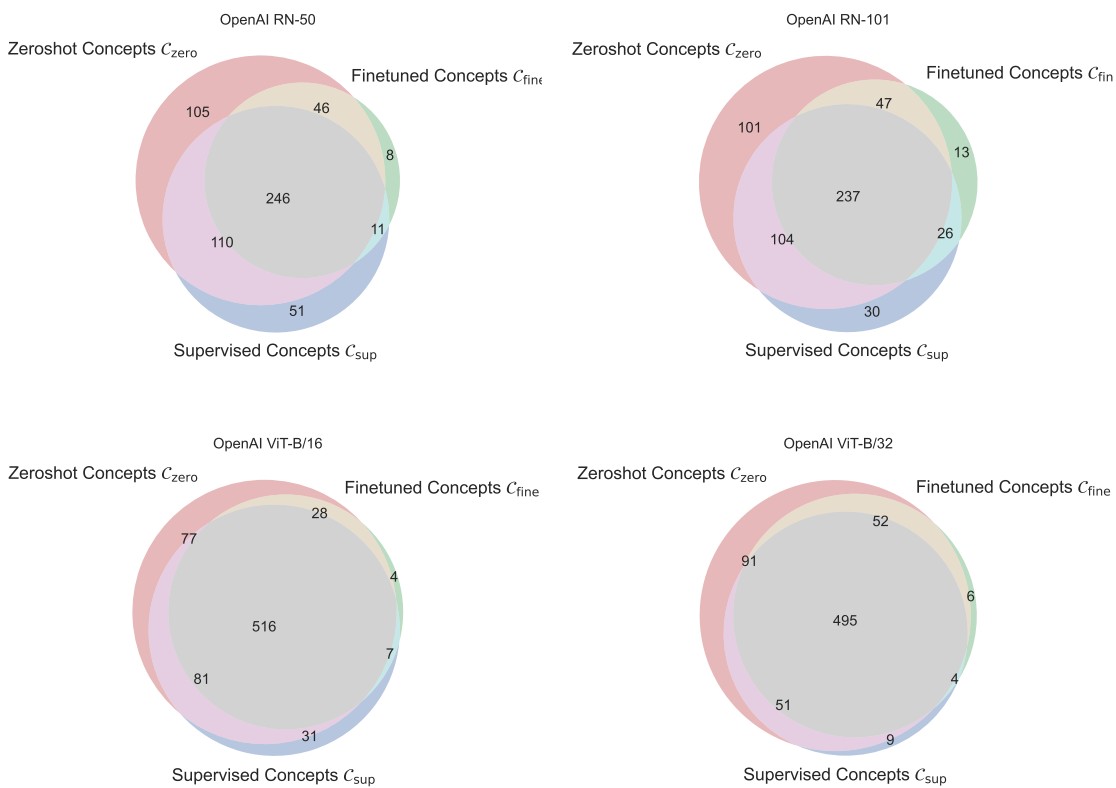

Figure 4: *Overlap between the concepts encoded in the representation space of different models for each OpenAI models. Zero-shot models encode many concepts not encoded other models.*

$|\{c \in [C] \mid \text{AP}_i^c \geq 0.9\}|$. The higher this number is, the more likely it is that the feature corresponding to $v_i$ is polysemantic. As a measure of polysemanticity for the model, we simply average this number over all singular vectors: $\text{polysemanticity} := d_H^{-1} \sum_{i=1}^{d_H} N_{\text{concept}}(i)$. By measuring this number for all zero-shot models, we found that this ranges between polysemanticity $= 3$ for the OpenAI ResNet 50 and polysemanticity $= 16$ for the LAION-2B ViT-B/16. By looking at the complete results in Appendix E.4, we also note that most CLIP models are on the higher side of this range, with typically more than 10 concepts per direction on average. Since the Broden dataset has no duplicate concepts, we deduce that these models are highly polysemantic.

> **Take-away 4.** A larger number of concepts encoded in a model's representation space is not an indication of robustness since the number of concepts encoded by some non-robust zero-shot ImageNet-Captions CLIP models can be as high as for some robust zero-shot CLIP models. However, both types of zero-shot CLIP models encode a higher number of concepts than their supervised counterparts, and among the ImageNet-Captions CLIP models, the amount of concepts varies depending on the type of language supervision used. This supports the idea that the amount of concepts encoded is related to the language supervision of CLIP models. The large number of concepts encoded in CLIP models makes these models polysemantic.

## 6  Related work

**Interpretability and CLIP models.** A number of works studied CLIP from a model-centric or interpretability perspective. We can broadly divide these works into two categories. (1) The first body of work, like ours, uses interpretability methods to gain a better understanding of CLIP. For instance, Li et al. (2022)

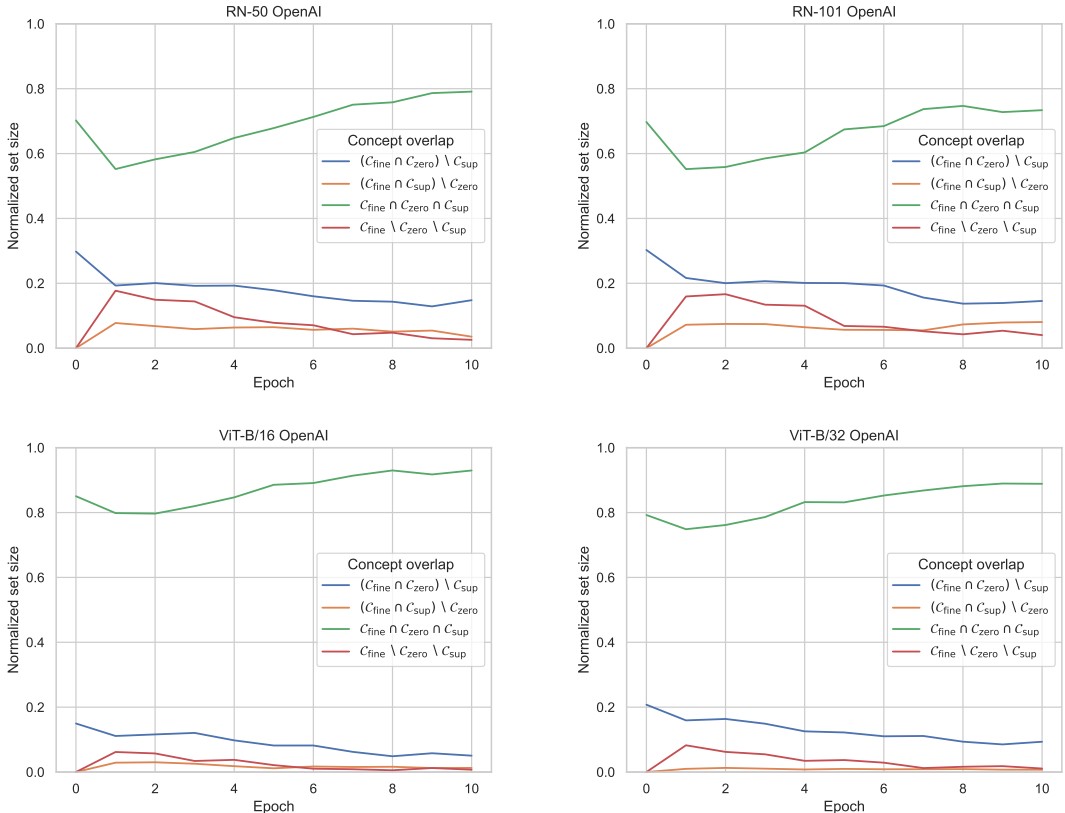

Figure 5: *Overlap of the finetuned model concepts with zero-shot and supervised models during finetuning, normalized at each epoch by the number of finetuned concepts $|\mathcal{C}_{\text{fine}}|$. The overlap with the zero-shot-only (i.e. not overlapping with supervised) concepts decreases (blue curve), while concepts shared with zero-shot and supervised models increases (green curve).*

analyze saliency maps of CLIP and found that they tend to focus on the background in images. Goh et al. (2021) analyzed CLIP ResNets and found *multimodal neurons*, that respond to the presence of a concept in many different settings. (2) The second body of work leverages CLIP to explain other models. For instance, Jain et al. (2022) use CLIP to label hard examples that are localized as a direction in any model's representation space. Similarly, Oikarinen & Weng (2022) use CLIP to label neurons by aligning their activation patterns with concept activation patterns on a probing set of examples. To the best of our knowledge, our work is the first to leverage interpretability to better understand the distribution shift robustness of CLIP.

**Outlier features in foundation models.** Outlier features in foundation models were first discovered in LLMs by Dettmers et al. (2022). Those features are found to have an adverse effect on the model quantization. The reason for which outlier features appear in LLMs is yet unknown. Elhage et al. (2023) investigated several possibles causes (such as layer normalization), but found no conclusive explanation. They conclude that the emergence of outlier features is most likely a relic of Adam optimization. Bondarenko et al. (2023) found that outlier features in transformers assign most of their mass to separator tokens and that modifying the attention mechanism (by clipping the softmax and using gated attention) decreases the amount of outlier features learned during pretraining. To the best of our knowledge, our work is the first to discuss outlier features outside of language and transformer models. We can also offer an alternative explanation for their emergence (see end of Section 4). Overall, our work shows that outlier features are more universal phenomena than previously known, and motivates further research to understand the mechanisms at play.

# 7 Discussion

The goal of this work was to generate a better understanding of what models that are robust to distribution shifts around ImageNet have learned from data that distinguishes them from non-robust models. To this end, we conducted a thorough investigation of the representation spaces of robust CLIP models and their non-robust counterparts, analyzing a total of 39 models.

We found outlier features (Dettmers et al., 2022; Elhage et al., 2023) to be an indication of robustness, since they can only be found in models that are robust to ImageNet distribution shifts. To the best of our knowledge, this is also the first time that outlier features were observed outside of language and transformer models. Since the presence of outlier features can be detected without access to the shifted datasets, we believe that they could be a useful tool for practitioners to get a feeling for the distribution shift robustness of a pretrained model during deployment, when the exact form of distribution shift is typically unknown. Interestingly, we can also validate this indicator on robust non-CLIP models (CoCa, Yu et al. (2022)) in Appendix D.

Lastly, we found that larger numbers of encoded concepts in the representation space are rather related to the type of language supervision than to be an indication of robustness, since they can also be found in non-robust ImageNet-Captions CLIP models and their number varies with the type of text used for language supervision. It would be interesting to further investigate this hypothesis by creating different types of language supervision for identical images, potentially leveraging state-of-the-art multimodal models, to create even richer signals.

In general, we believe an interesting avenue for future research would be to extend the analysis of this work to dataset shifts beyond the ImageNet family, to see if our analysis remains relevant beyond the much investigated ImageNet shifts.

### Acknowledgments

The authors thank Hadi Pour Ansari, Dan Busbridge, Miguel Sarabia del Castillo, Barry Theobald, and Nicholas Apostoloff for their helpful feedback and their support.

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

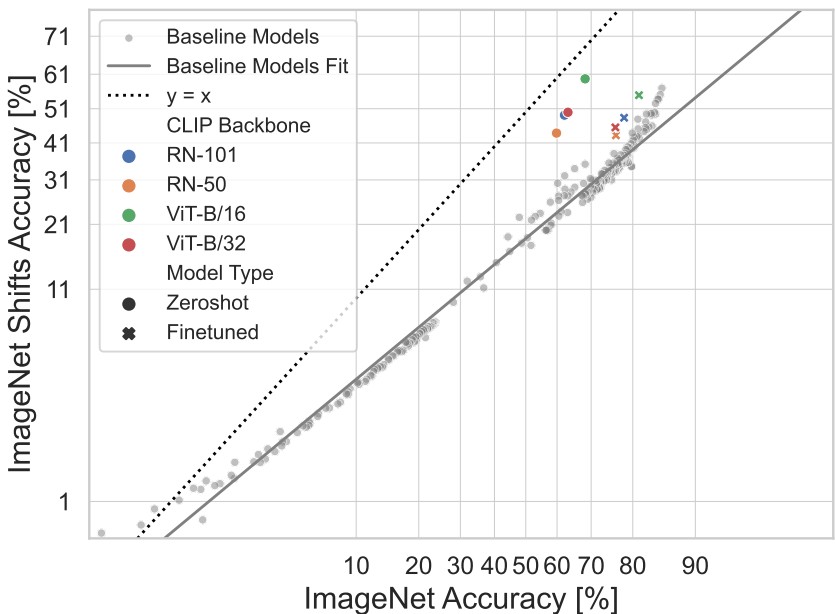

Figure 6: *Accuracies of baseline models and OpenAI CLIP models on ImageNet and (average) accuracies on its five natural shifts (ImageNet-V2, ImageNet-R, ImageNet-Sketch, ImageNet-A, ObjectNet). The zero-shot OpenAI CLIP models accuracies are substantially above the baseline fit, i.e., they have high ER. The fine-tuned models are closer to the baseline fit, i.e., they have lower, but still significant ER.*

## A   Computing Effective Robustness

In this section, we define *Effective Robustness* (ER).

**Context.** ER has emerged as a natural metric to measure how the performance of a model on a reference distribution (in-distribution) generalizes to natural shifts of this distribution (Fang et al., 2022). When plotting the in-distribution accuracy (X-axis, *logit scaling*) against the average shifted-distribution accuracy (Y-axis, *logit scaling*) of various architectures trained on ImageNet, Taori et al. (2020) found that most of the existing models lie on the same line. They also found that models trained with *substantially* more data lie above this line, showing a desirable gain in shifted-distribution accuracy for a fixed in-distribution accuracy. They coined this vertical lift above the line as *Effective Robustness.*

**Computing ER.** To quantify ER, following Taori et al. (2020), one gathers the ImageNet test accuracy $\texttt{ACC}(I)$ and the average accuracy over the ImageNet shifts $\texttt{ACC}(S)$ of a set of reference models trained on ImageNet and fits a linear model on this pool of accuracies to map $\text{logit}[\texttt{ACC}(I)]$ to $\text{logit}[\texttt{ACC}(S)]$, with the logit function $\text{logit} : [0,1] \to \mathbb{R}$ defined as $x \mapsto \ln(x) - \ln(1-x)$. The resulting line can be used to predict what (logit) accuracy we would expect to see on the ImageNet shifts, given a (logit) accuracy on the original ImageNet. Given a new model that has accuracy $\texttt{ACC}(I)$ on ImageNet and average accuracy $\texttt{ACC}(S)$ on the canonical ImageNet shifts, $\texttt{ER}$ is computed as:

$$\texttt{ER}(\texttt{ACC}(S), \texttt{ACC}(I)) :=$$
$$\texttt{ACC}(S) - \text{logit}^{-1}\left[\beta_1 \text{logit}\left[\texttt{ACC}(I)\right] + \beta_0\right]. \tag{5}$$

By fitting a line on the baseline accuracies collected by Taori et al. (2020), we get a slope of $\beta_1 = .76$ and an intercept of $\beta_0 = -1.49$, with a Pearson correlation $r = .99$. This line, along with the baseline models, can be observed in Figure 6.

# B  Model Accuracies on ImageNet and its shifts

In Table 1 we show the top-1 accuracies that the models analysed in the main part of the paper achieve on the ImageNet test set, and in Table 2 the test accuracies on the five shifted datasets (averaged).

Table 1: *ImageNet test set accuracies for the models under investigation*

| Backbone | Pretraining data | Zero-shot CLIP | Finetuned CLIP | ImageNet supervised |
|---|---|---|---|---|
| ResNet50 | OpenAI | 60 % | 76 % | 70 % |
| | YFCC-15M | 32 % | 69 % | |
| | CC-12M | 36 % | 69 % | |
| | ImageNet-Captions-t | 25 % | | |
| | ImageNet-Captions-td | 27 % | N.A. | |
| | ImageNet-Captions-tdt | 28 % | | |
| ResNet101 | OpenAI | 62 % | 78 % | 71 % |
| | YFCC-15M | 34 % | 72 % | |
| ViT-B-16 | OpenAI | 68 % | 81 % | 80 % |
| | LAION-400M | 67 % | 80 % | |
| | LAION-2B | 70 % | 81 % | |
| | DataComp | 63 % | 78 % | |
| ViT-B-32 | OpenAI | 63 % | 78 % | 75 % |
| | LAION-400M | 60 % | 76 % | |
| | LAION-2B | 66 % | 76 % | |
| ViT-L-14 | OpenAI | 75 % | 85 % | N.A. |
| | LAION-400M | 73 % | 84 % | |
| | LAION-2B | 74 % | 84 % | |
| | DataComp | 79 % | 85 % | |

Table 2: *Average test set accuracies on ImageNet-V2, ImageNet-R, ImageNet-Sketch, ImageNet-A, ObjectNet for the models under investigation.*

| Backbone | Pretraining data | Zero-shot CLIP | Finetuned CLIP | ImageNet supervised |
|---|---|---|---|---|
| ResNet50 | OpenAI | 44 % | 43 % | 30 % |
| | YFCC-15M | 18 % | 30 % | |
| | CC-12M | 27 % | 36 % | |
| | ImageNet-Captions-t | 9 % | | |
| | ImageNet-Captions-td | 9 % | N.A. | |
| | ImageNet-Captions-tdt | 10 % | | |
| ResNet101 | OpenAI | 49 % | 48 % | 31 % |
| | YFCC-15M | 21 % | 33 % | |
| ViT-B-16 | OpenAI | 60 % | 55 % | 41 % |
| | LAION-400M | 56 % | 56 % | |
| | LAION-2B | 60 % | 58 % | |
| | DataComp | 52 % | 53 % | |
| ViT-B-32 | OpenAI | 50 % | 45 % | 35 % |
| | LAION-400M | 48 % | 48 % | |
| | LAION-2B | 54 % | 48 % | |
| ViT-L-14 | OpenAI | 72 % | 67 % | N.A. |
| | LAION-400M | 64 % | 64 % | |
| | LAION-2B | 66 % | 66 % | |
| | DataComp | 76 % | 71 % | |

# C  Zero-shot and finetuned models' differences are localized

In this appendix, we apply central kernel alignment (CKA) to identify where changes between robust zero-shot CLIP models and their less robust fine-tuned counterparts occur. Kornblith et al. (2019) introduce

the CKA metric to quantify the degree of similarity between the activation patterns of two neural network layers. It takes two batch of activation vectors $a$ and $b$, it computes their normalized similarity in terms of the Hilbert-Schmidt Independence Criterion (HSIC, Gretton et al. (2005)):

$$\text{CKA}(a, b) := \frac{\text{HSIC}(a, b)}{\sqrt{\text{HSIC}(a, a)}\sqrt{\text{HSIC}(b, b)}}$$

We use the PyTorch-Model-Compare package (Subramanian, 2021) to compute this metric between the activation vectors of zero-shot models and their finetuned counterparts for each layer in the backbone. The results are shown in Figure 7 and Figure 8. Across architectures and pretraining sets, we find that there is often a large drop in CKA between zero-shot and finetuned models occurring in the last layer. This makes the activations in the last layer a particularly interesting layer to analyse when investigating ER, as fine-tuned models typically have only half the ER of their zero-shot counterpart (see Figure 1a).

## D  Robustness indicators in non-CLIP models

In addition to the CLIP models investigated in the main paper, we below investigate CoCa models (Yu et al., 2022) pre-trained on LAION-2B as another set of robust multimodal models which do not fall into the CLIP family. We see that our findings extend to these non-CLIP multimodal models as well.

First, in Table 3, we confirm that these models have high effective robustness when used as zero-shot classifiers. As for the other models in the main part of the paper, we observe that finetuning on ImageNet decreases the effective robustness of these classifiers.

Table 3: *ER for the models of the CoCa family, as calculated by Equation (5) (accuracies on ImageNet shown in brackets). We see that also for these models, ER decreases from Zero-shot to Finetuned.*

| Backbone | CoCa Zero-shot | CoCa Finetuned |
|---|---|---|
| ViT-B-32 | 24% (64%) | 14% (76%) |
| ViT-L-14 | 34% (76%) | 21% (84%) |

Next, in Table 4, we show that all the zero-shot models have high kurtosis, which implies the existence of outlier features in their representation space. Additionally, we show that finetuning again decreases the kurtosis.

Table 4: *Results of the kurtosis analysis showing outlier features present also in robust models with ViT-L-14 backbone or of CoCa family, but disappearing as soon as models are finetuned. Values calculated according to Equation (1) over all ImageNet test examples.*

| Backbone | CoCa Zero-shot | CoCa Finetuned |
|---|---|---|
| ViT-B-32 | 12.0 | 3.6 |
| ViT-L-14 | 15.5 | 4.6 |

Finally, in Table 5, we see that zero-shot model encodes more concepts. Again, we see that finetuning removes some concepts from the model's representation space. However, from our experiments with ImageNet-Captions CLIP models, we know that this does not correspond to a robustness indicator.

Table 5: *Results of the unique concept analysis, showing total # of unique Broden concepts encoded in last layers, as in Equation (4). Zero-shot models encode substantially more concepts.*

| Backbone | CoCa Zero-shot | CoCa Finetuned |
|---|---|---|
| ViT-B-32 | 674 | 530 |
| ViT-L-14 | 747 | 629 |

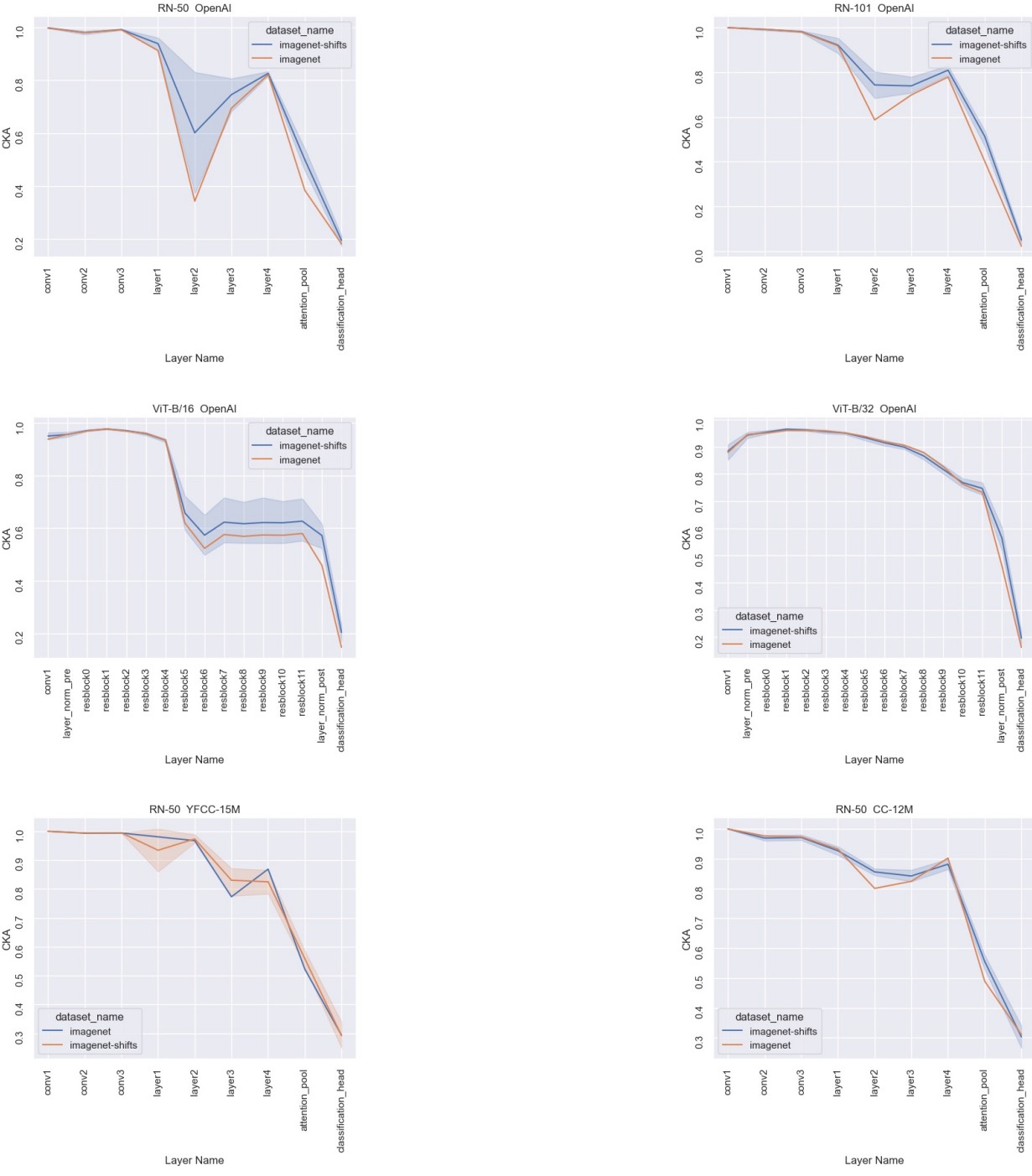

Figure 7: *Result of layer by layer CKA comparison between zero-shot CLIP and its counterpart that was fine-tuned on ImageNet for various backbones and pretraining sets (Part 1). In orange, CKA between activation vectors on ImageNet test set. In blue, CKA between activation vectors on shifted ImageNet sets (average as solid line, standard deviation in shaded blue). Typically, we see large drops of CKA in the last layer.*

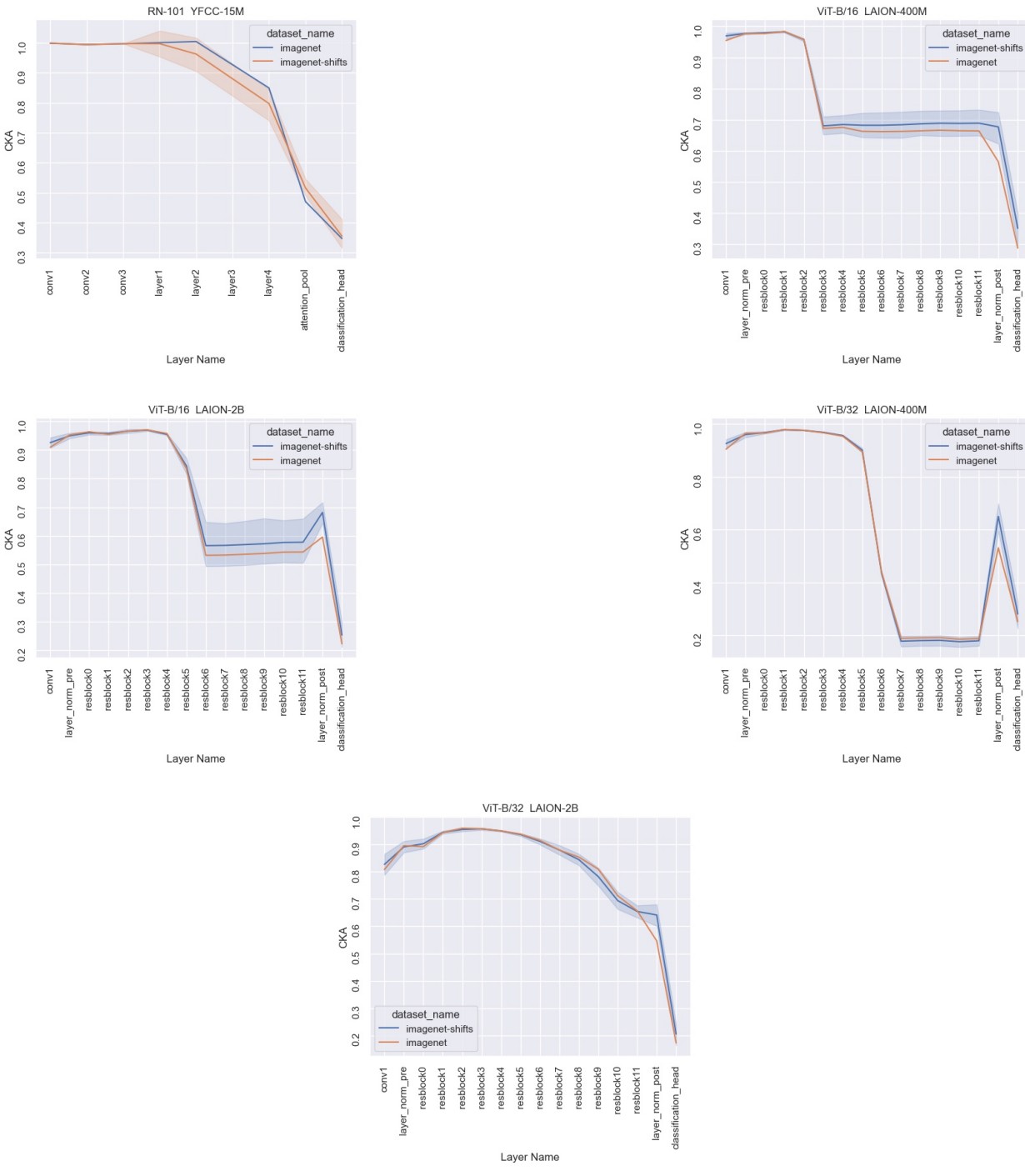

Figure 8: *Result of layer by layer CKA comparison between zero-shot CLIP and its counterpart that was fine-tuned on ImageNet for various backbones and pretraining sets (Part 2). In orange, CKA between activation vectors on ImageNet test set. In blue, CKA between activation vectors on shifted ImageNet sets (average as solid line, standard deviation in shaded blue). Typically, we see large drops of CKA in the last layer.*

# E  Further experiment results

## E.1  Importance score analysis for each model

In Figure 9 and Figure 10 we show the the full distribution of importance scores (Equation (2)) over all representation space directions for all models.

We see that robust zero-shot CLIP models and finetuned CLIP models have one strongly privileged direction, typically orders of magnitude larger than the bulk of importance scores. For the non-robust models trained on ImageNet and ImageNet-Caption, this single outlier value in importance scores does not exist.

## E.2  Pruning analysis for each model

Figure 11 and Figure 12 show the effect of gradually pruning the least important SV of $W$ on ER and ACC for all models that were not pretrained on the OpenAI pretraining dataset, similar to the analysis shown in Figure 13 in the main paper.

Qualitatively, our finding from the main paper is confirmed across the remaining models we investigate: A small subset (typically around 20%) of privileged directions in representation space explain the high ER of zero-shot models. The remaining directions can be pruned without significantly impacting neither ER nor ACC.

## E.3  Top-3 concepts in most dominant direction for each model

We look at the concepts with highest AP in the most privileged direction of each model represented in Figure 9, similar to what we did in Section 5 for the most privileged direction of each model in Figure 2b. The results are shown in Table 6. Again, we observe that for the majority of cases, the robust zero-shot models encode concepts related to textures as their top concepts encoded along the privileged directions (e.g., *scaly*, *meshed*, or *matted*), while the less robust finetuned models encode more concrete concepts (e.g., *carrousel*, *book stand*, or *pantry*).

Table 6: *Top-3 concepts with highest AP encoded in the most privileged direction of each model. For each concept, the AP is included in brackets.*

| | | Top-1 Concept | | Top-2 Concept | | Top-3 Concept | |
|---|---|---|---|---|---|---|---|
| Backbone | Step Pretraining | Finetuned | Zeroshot | Finetuned | Zeroshot | Finetuned | Zeroshot |
| RN-101 | OpenAI | flight of stairs natural s ( 0.9) | moon bounce s ( 0.99) | movie theater indoor s ( 0.88) | inflatable bounce game ( 0.99) | home theater s ( 0.86) | ball pit s ( 0.91) |
| | Supervised ImageNet | auto mechanics indoor s ( 0.96) | N.A. | labyrinth ( 0.94) | N.A. | hay ( 0.94) | N.A. |
| | YFCC-15M | chapel s ( 1) | ice cream parlor s ( 0.99) | pantry s ( 0.97) | temple ( 0.98) | pantry ( 0.97) | temple east asia s ( 0.95) |
| RN-50 | CC-12M | sacristy s ( 0.98) | wheat field s ( 0.98) | funeral chapel s ( 0.97) | meshed ( 0.98) | formal garden s ( 0.96) | polka dotted ( 0.96) |
| | OpenAI | mountain pass ( 0.99) | knitted ( 0.95) | butte s ( 0.94) | chequered ( 0.91) | water mill s ( 0.89) | wheat field s ( 0.87) |
| | Supervised ImageNet | kiosk indoor s ( 0.95) | N.A. | vegetable garden s ( 0.88) | N.A. | sacristy s ( 0.86) | N.A. |
| | YFCC-15M | liquor store indoor s ( 0.97) | polka dotted ( 0.98) | book stand ( 0.96) | lined ( 0.97) | horse drawn carriage ( 0.89) | dotted ( 0.97) |
| ViT-B/16 | DataComp | carrousel s ( 0.98) | jail cell s ( 0.84) | banquet hall s ( 0.93) | gift shop s ( 0.83) | carport freestanding s ( 0.89) | manhole s ( 0.82) |
| | LAION-2B | flood s ( 0.9) | stained ( 0.89) | catwalk s ( 0.87) | scaly ( 0.86) | rubble ( 0.85) | cracked ( 0.85) |
| | LAION-400M | book stand ( 0.97) | temple ( 0.97) | bookstore s ( 0.94) | courtyard s ( 0.95) | rudder ( 0.93) | cabana s ( 0.94) |
| | OpenAI | martial arts gym s ( 0.95) | meshed ( 0.92) | jail cell s ( 0.92) | flecked ( 0.92) | throne room s ( 0.91) | perforated ( 0.92) |
| | Supervised ImageNet | hot tub outdoor s ( 0.99) | N.A. | bedchamber s ( 0.98) | N.A. | stadium baseball s ( 0.97) | N.A. |
| | YFCC-15M | fountain s ( 0.68) | N.A. | black c ( 0.67) | N.A. | air base s ( 0.62) | N.A. |
| ViT-B/32 | DataComp | zen garden s ( 0.95) | ice cream parlor s ( 0.67) | dolmen s ( 0.94) | bullring ( 0.67) | gift shop s ( 0.88) | junkyard s ( 0.67) |
| | LAION-2B | viaduct ( 0.93) | stained ( 0.91) | cargo container interior s ( 0.91) | scaly ( 0.91) | labyrinth ( 0.9) | matted ( 0.88) |
| | LAION-400M | barnyard s ( 0.86) | scaly ( 0.94) | subway interior s ( 0.86) | jail cell s ( 0.9) | bird feeder ( 0.86) | manhole s ( 0.9) |
| | OpenAI | tennis court ( 1) | meshed ( 0.95) | batters box s ( 0.98) | perforated ( 0.93) | kennel indoor s ( 0.93) | flecked ( 0.91) |
| | Supervised ImageNet | television studio s ( 0.88) | N.A. | barbecue ( 0.87) | N.A. | lined ( 0.85) | N.A. |

## E.4  Polysemanticty for each model

We report the polysemanticity metric computed as per Section 5 for all zero-shot models in Table 7. As claimed in the paper, this ranges from polysemanticity = 3 for the OpenAI ResNet 50 to polysemanticity = 16 for the LAION-2B ViT-B/16, with typically more than 10 concepts encoded in one direction on average.

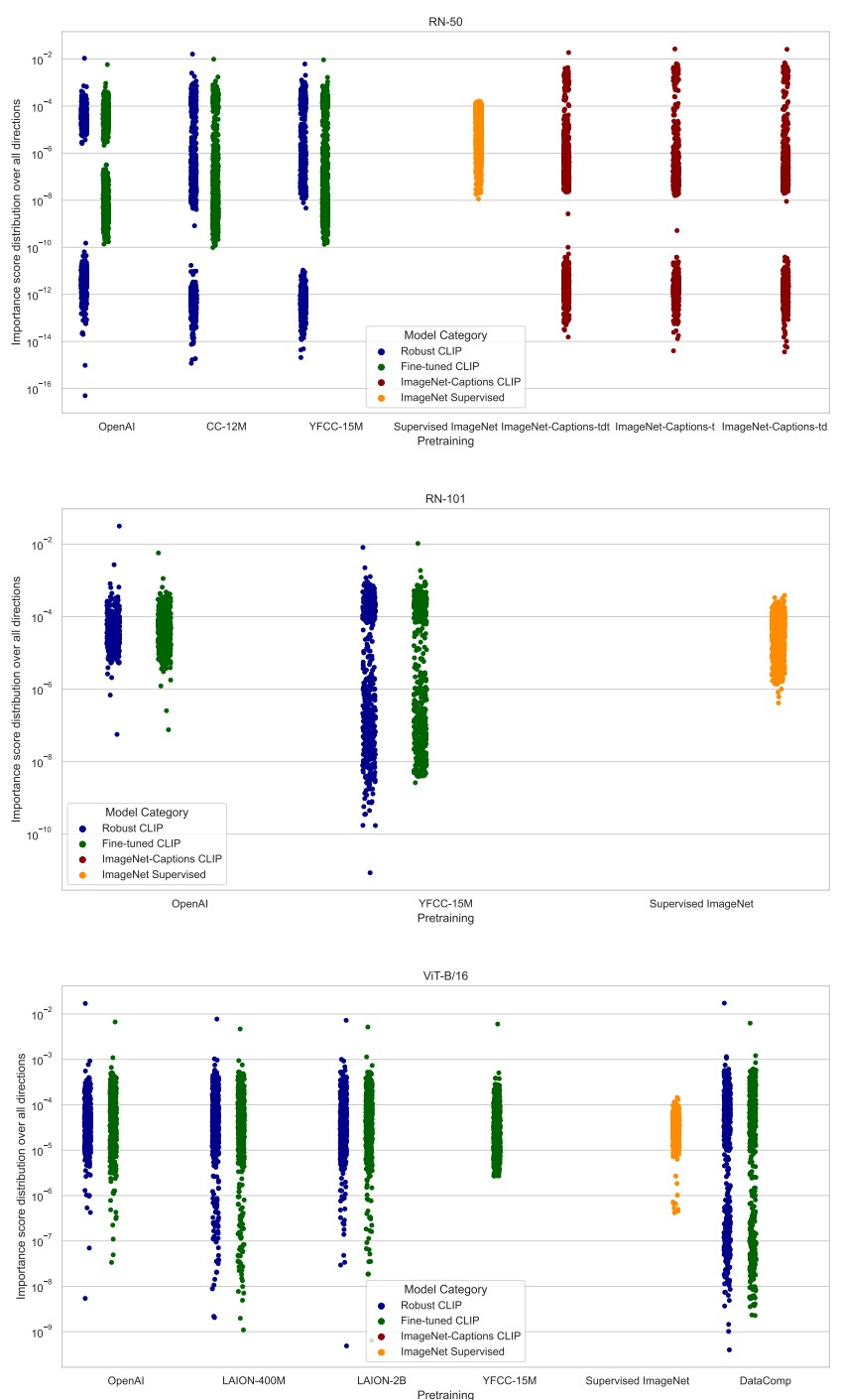

Figure 9: Distribution of importance over all RSVs in the representation space for all models with ResNet50, ResNet101, and ViT-B-16 backbones. Robust zero-shot CLIP models (blue) have one strongly privileged direction. Fine-tuned models (green) still exhibit one strong privileged direction, but with lower importance than the robust zero-shot models. The supervised models (orange) do not have them at all. The non-robust ImageNet-Caption models (red) do have a few directions that are a little bit larger than the bulk, but not separated by orders of magnitude like for the robust and fine-tuned CLIP models.

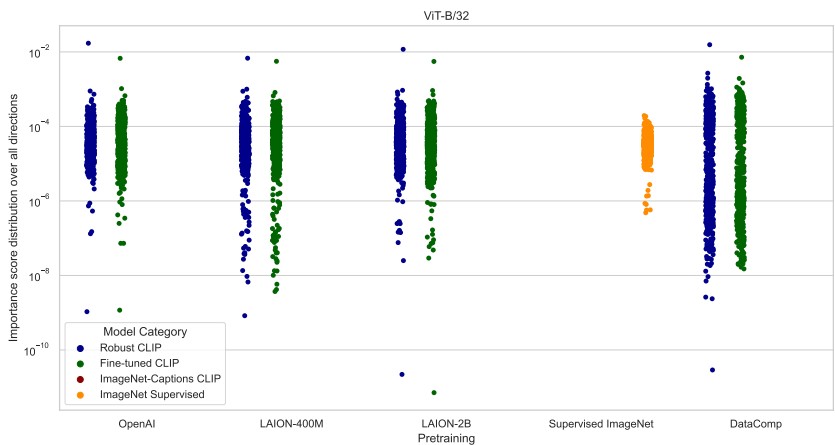

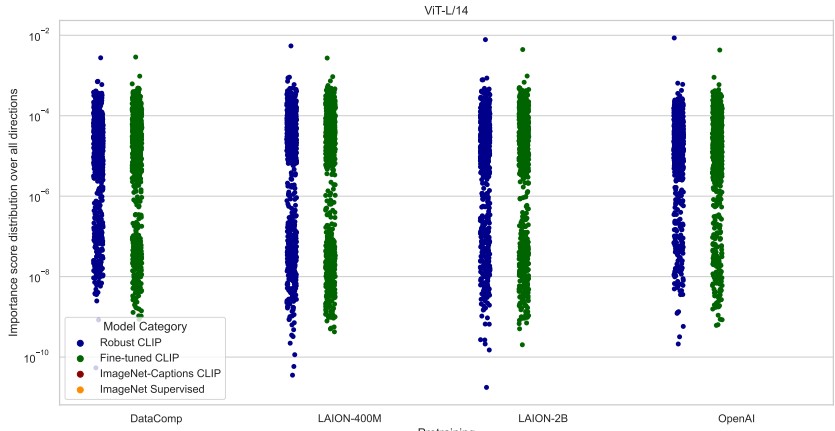

Figure 10: Distribution of importance over all RSVs in the representation space for all models with ViT-B-32 and ViT-L-14 backbones. Robust zero-shot CLIP models (blue) have one strongly privileged direction. Fine-tuned models (green) still exhibit one strong privileged direction, but with lower importance than the robust zero-shot models. The supervised models (orange) do not have them at all.

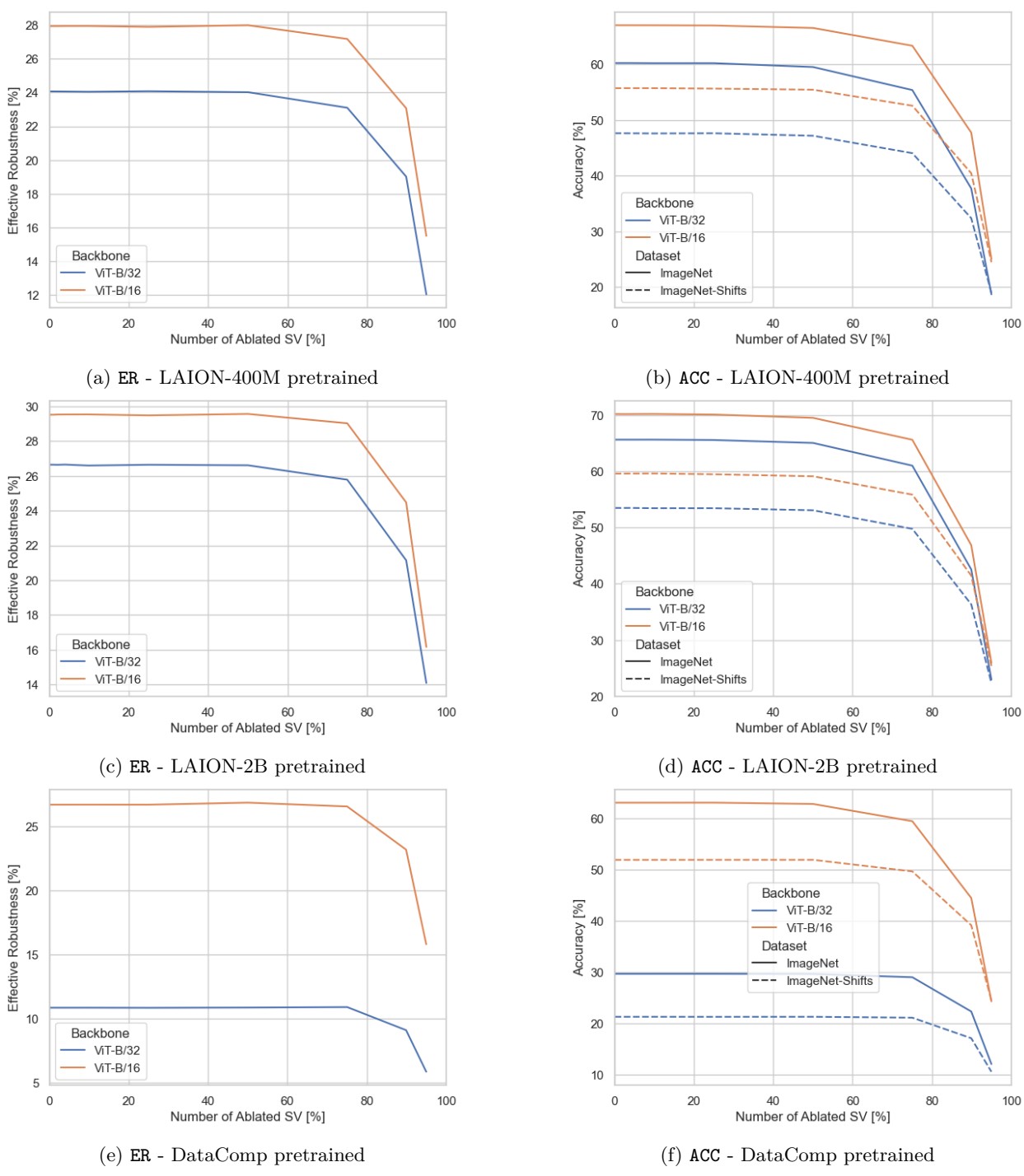

(a) ER - LAION-400M pretrained

(b) ACC - LAION-400M pretrained

(c) ER - LAION-2B pretrained

(d) ACC - LAION-2B pretrained

(e) ER - DataComp pretrained

(f) ACC - DataComp pretrained

Figure 11: *Effect of gradually pruning the least important SV of W on ER and ACC for LAION-400M, LAION-2B, and DataComp pretrained models.*

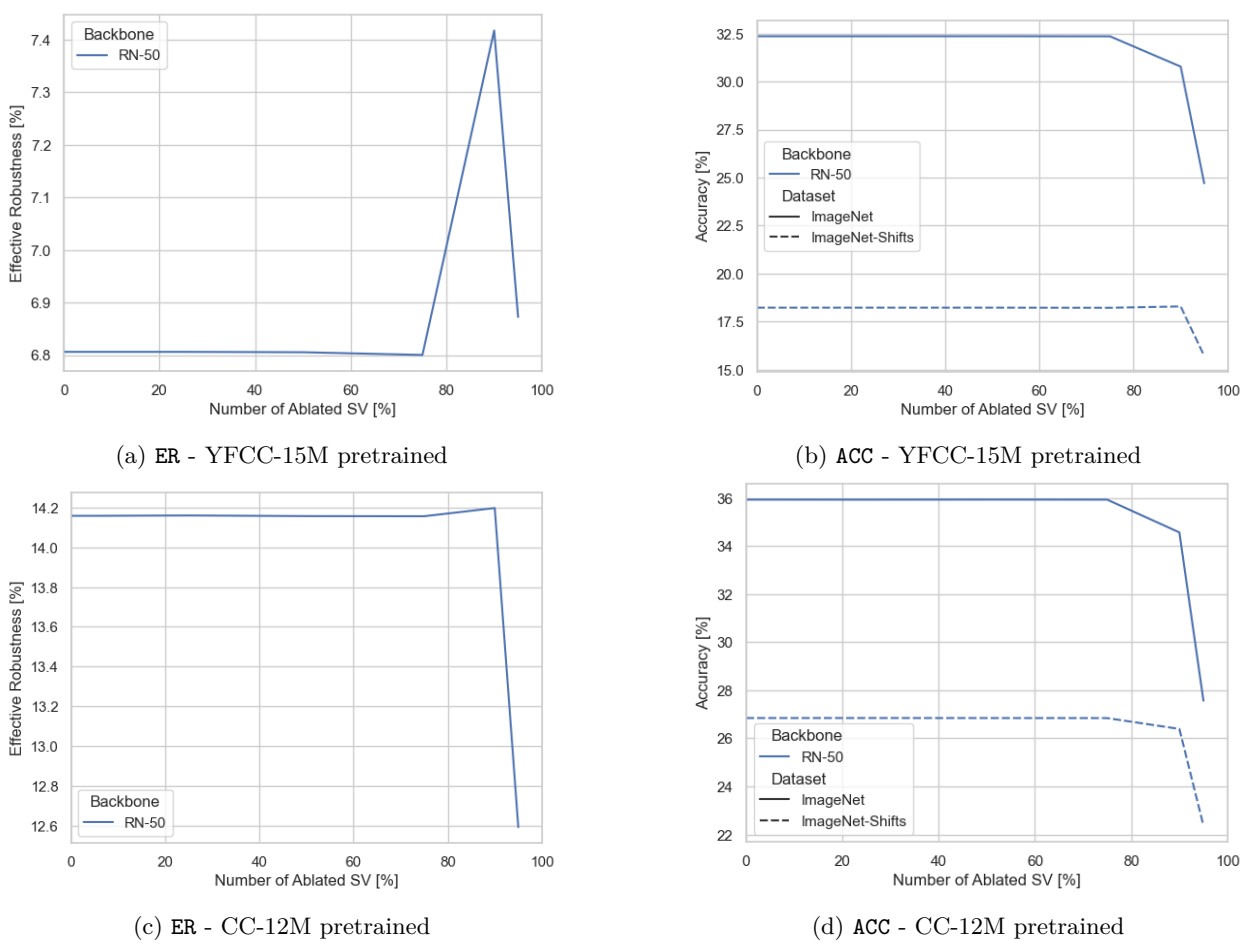

(a) ER - YFCC-15M pretrained

(b) ACC - YFCC-15M pretrained

(c) ER - CC-12M pretrained

(d) ACC - CC-12M pretrained

Figure 12: *Effect of gradually pruning the least important SV of W on ER and ACC for YFCC-15M and CC-12M pretrained models.*

Table 7: *Polysemanticity of zero-shot CLIP models, showing average # of concepts encoded in RSV of last layer (i.e. with* `AP` $\geq 0.9$ *on Broden dataset concepts).*

| Backbone | Pretraining data | Polysemanticity |
|---|---|---|
| ResNet50 | OpenAI | 3.0 |
| | YFCC-15M | 6.4 |
| | CC-12M | 5.3 |
| | ImageNet-Captions-t | 10.5 |
| | ImageNet-Captions-td | 10.5 |
| | ImageNet-Captions-tdt | 4.5 |
| ResNet101 | OpenAI | 3.5 |
| | YFCC-15M | 7.8 |
| ViT-B-16 | OpenAI | 14.5 |
| | LAION-400M | 11.9 |
| | LAION-2B | 16.0 |
| | DataComp | 14.1 |
| ViT-B-32 | OpenAI | 14.1 |
| | LAION-400M | 11.5 |
| | LAION-2B | 11.1 |

## F   Intuition behind generalization of outlier features

Below, we give more details and an intuition why outlier features can be generalized from the canonical basis $\{e_1, \ldots, e_{d_H}\}$ (we can write $h_i = \mathrm{Proj}_{e_i}(h)$) to be any set of directions of the representation space that receive a projection substantially above average:

Let us assume, for instance, that two of the elements in the canonical basis $e_1$ and $e_2$ correspond to outlier features. This means that an activation vector $h$ related to an input image $x$ has projections $h_1 = \mathrm{Proj}_{e_1}(h)$ and $h_2 = \mathrm{Proj}_{e_2}(h)$ substantially above the average $h_1, h_2 \gg n^{-1} \sum_{i=1}^{n} h_i$. Now let us define a new unit vector $e_1' = 2^{-1/2}(e_1 + e_2)$. We deduce that the projection onto this vector is also substantially higher than average $h_1' = \mathrm{Proj}_{e_1'}(h) = 2^{-1/2}(h_1 + h_2) \gg n^{-1} \sum_{i=1}^{n} h_i$. Hence, the unit vector $e_1'$ can be considered as an outlier feature in a new non-canonical basis. In general, we can extend the notion of *outlier features* to any vector in the span$\{e_1, \ldots, e_n \in \mathbb{R}^n\}$.

## G   Pruning results

**Pruning non-privileged directions.** Given that we have established that outlier features induce privileged directions in representation space, it seems interesting to check their role in model performance. To that aim, we gradually prune each RSV $v_i$ by increasing order of $\sigma_i$ by setting $\sigma_i \leftarrow 0$ in the singular value expansion[4] $W = \sum_{i=1}^{\mathrm{rank}(W)} \sigma_i u_i v_i^{\intercal}$. By pruning a variable proportion of the singular vectors, we obtain the results in Figure 13. We see that the 80% least important RSV of the representation space can be pruned without a substantial effect on performance, i.e. that the robust models are low-rank in their last layer where they have privileged directions.

When extending the pruning experiment to finetuned CLIP models and supervised models trained only with ImageNet, we make the two interesting observations from these new results (see Figure 14):

**All models are low-rank.** For all the models (zero-shot, finetuned and supervised), the performances are not substantially affected if we remove the 80% least important singular directions of their representation space (compare to Table 1). This shows that many existing models admit good low-rank approximations. This also demonstrates that the fact that these models are low-rank is not necessarily an indicator of robustness.

---

[4]Note that sorting the RSV $v_i$ by increasing $\sigma_i$ is similar to sorting the RSV by increasing Importance($i$), since these two variables are related by a Spearman rank correlation $\rho = 96\%$

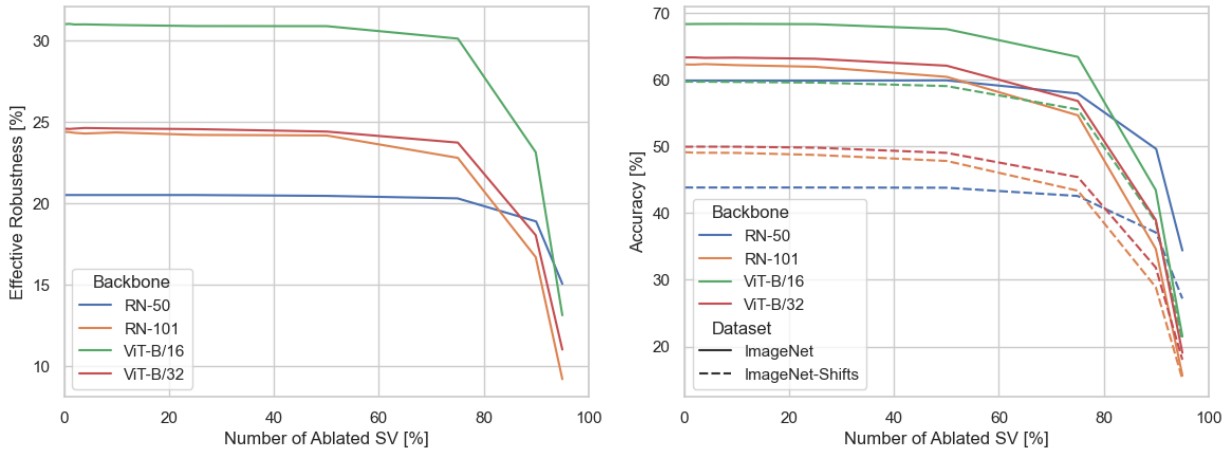

Figure 13: *Effect of gradually pruning the least important SV of W on ER and ACC.* The least 80% important SV can be pruned without any substantial effect.

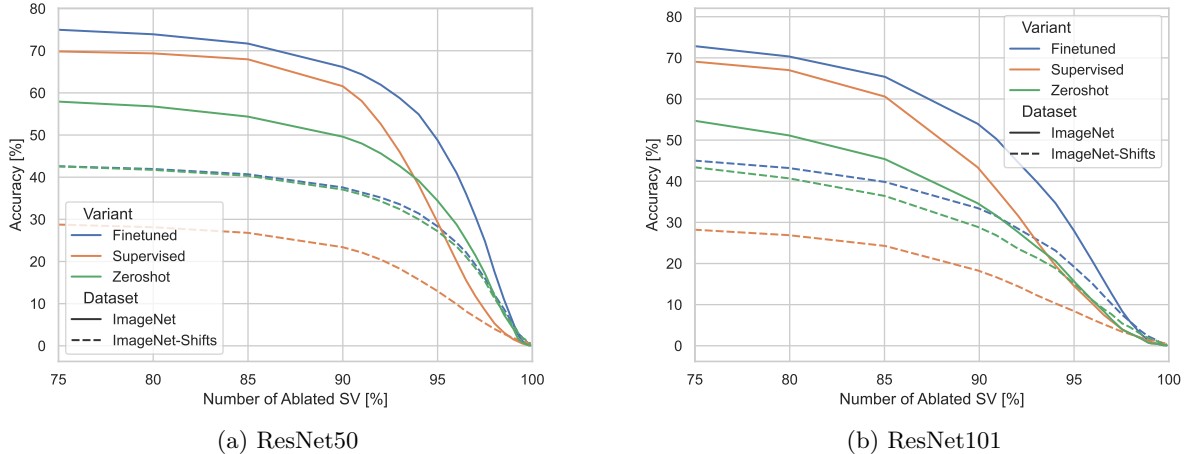

Figure 14: *Extension of pruning results to finetuned and ImageNet supervised models. Zero-shot models obtained with OpenAI pretraining set.*

**Faster drop for supervised models.** When the number of ablated singular values ranges between $80\% - 100\%$, we see that the ImageNet accuracy of supervised models drop substantially faster than the accuracy of the finetuned and the zero-shot models. In fact, for the ResNet50, the ImageNet accuracy curves even cross. This implies that the most important direction of the zero-shot model's representation space better discriminate between ImageNet classes than the most important directions of the supervised model's representation space. In the former case, these directions correspond to the zero-shot model's privileged directions. We believe that this new result further reinforces the importance of privileged directions to understand the performances of robust models.

## H    Details on experiments

### H.1    Finetuned CLIP models

To obtain the finetuned CLIP models, we proceed as follows. We start from building the zero-shot CLIP models as described in Section 3. As Wortsman et al. (2022b), we then finetune these models for 10 epochs

on the ImageNet training set, using a batch size of 256 and a learning rate of $3 \cdot 10^{-5}$ with a cosine annealing learning rate scheduler and a warm-up of 500 steps. We use the AdamW optimizer and set the weight decay to 0.1.

### H.2 Supervised ImageNet models

We note that the ResNets used by Radford et al. (2021) have small modifications, such as the usage of attention pooling. Unfortunately, we are not aware of any public weights for such modified architectures trained on ImageNet from scratch. We thus train these modified ResNet models from scratch for 90 epochs on the ImageNet training set, using a batch size of 1024. We use AdamW, and a learning rate schedule decaying from $10^{-3}$ to $10^{-4}$ after 30 epochs and to $10^{-5}$ after 60 epochs (with a warm-up period of 5,000 steps). We set weight decay to $10^{-2}$. We use the standard augmentations of horizontal flip with random crop as well as label smoothing.

For the ViT models, loadable checkpoints with identical architectures were available from torchvision (TorchVision maintainers and contributors, 2016), and we thus use those directly.

## I Analysis of Wise-FT models

In this appendix, we use the approach of Wise-FT (Wortsman et al., 2022b) to obtain a continuous spectrum of ER. Given a zero-shot model $f_{\theta_0}$ with weights $\theta_0 \in \Theta$ and a finetuned model $f_{\theta_1}$ with weights $\theta_1 \in \Theta$, Wortsman et al. (2022b) propose to interpolate between the two models in weight space. This is done by taking a combination $\theta_\alpha := (1 - \alpha) \cdot \theta_0 + \alpha \cdot \theta_1$ for some interpolation parameter $\alpha \in [0, 1]$. One then defines a new model $f_{\theta_\alpha}$ based on the interpolated weights.

Surprisingly, interpolating between zero-shot CLIP models and finetuned CLIP models produce models with good performances. To illustrate that, we perform the Wise-FT interpolation with all the models from our pool. We report the ImageNet & shift accuracies of these models in Figures 15 and 16. For the OpenAI and LAION models in Figure 15, we observe that the shift accuracy of interpolated models often surpass both the zero-shot and the finetuned models. The YFCC-15M and CC-12M models in Figure 16 exhibit a different trend: both ImageNet & shift accuracies increase monotonically as $\alpha$ sweeps from zero-shot to finetuned. This is likely due to the low accuracy of the corresponding zero-shot models.

By analyzing the ER of interpolated OpenAI and LAION models in Figure 17, we see that the ER gradually degrades as $\alpha$ sweeps between the zero-shot and the finetuned models. Interestingly, the ER of YFCC-15M and CC-12M models in Figure 18 peaks at $\alpha = .4$ and then decreases monotonically.

Let us now look at how our robustness indicators evolve as we sweep $\alpha$ between zero-shot and finetuned models. Ideally, if these indicators are good ER proxies, they should exhibit similar trends as the ones described in the previous paragraph. For the OpenAI and LAION models, we indeed observe in Figures 19 and 21 that the kurtosis and the number of unique encoded concepts gradually decrease as $\alpha$ sweeps from zero-shot to finetuned models. Similarly, we observe in Figures 20 and 22 that these two metrics start to substantially after $\alpha = 0.4$ for the YFCC-15M and CC-12M models. This suggests that these two metrics constitute a good proxy to track how the ER of a given model evolves.

Note that the Wise-FT idea has since been generalized to a combination of several finetuned models by Wortsman et al. (2022a) with *model soups*. We leave the investigations of model soups for future work.

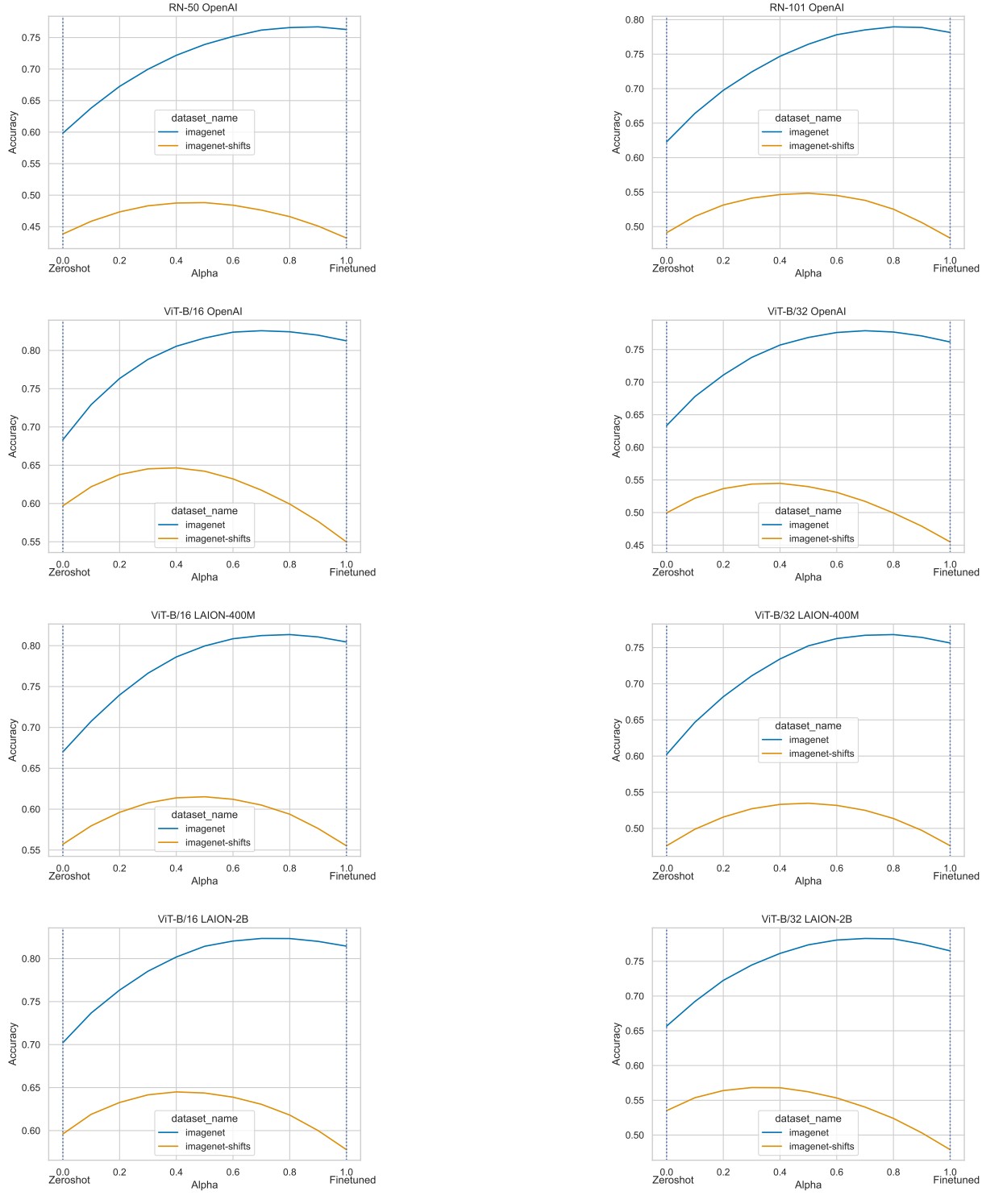

Figure 15: *Accuracies on ImageNet & shifts for Wise-FT models (1/2).*

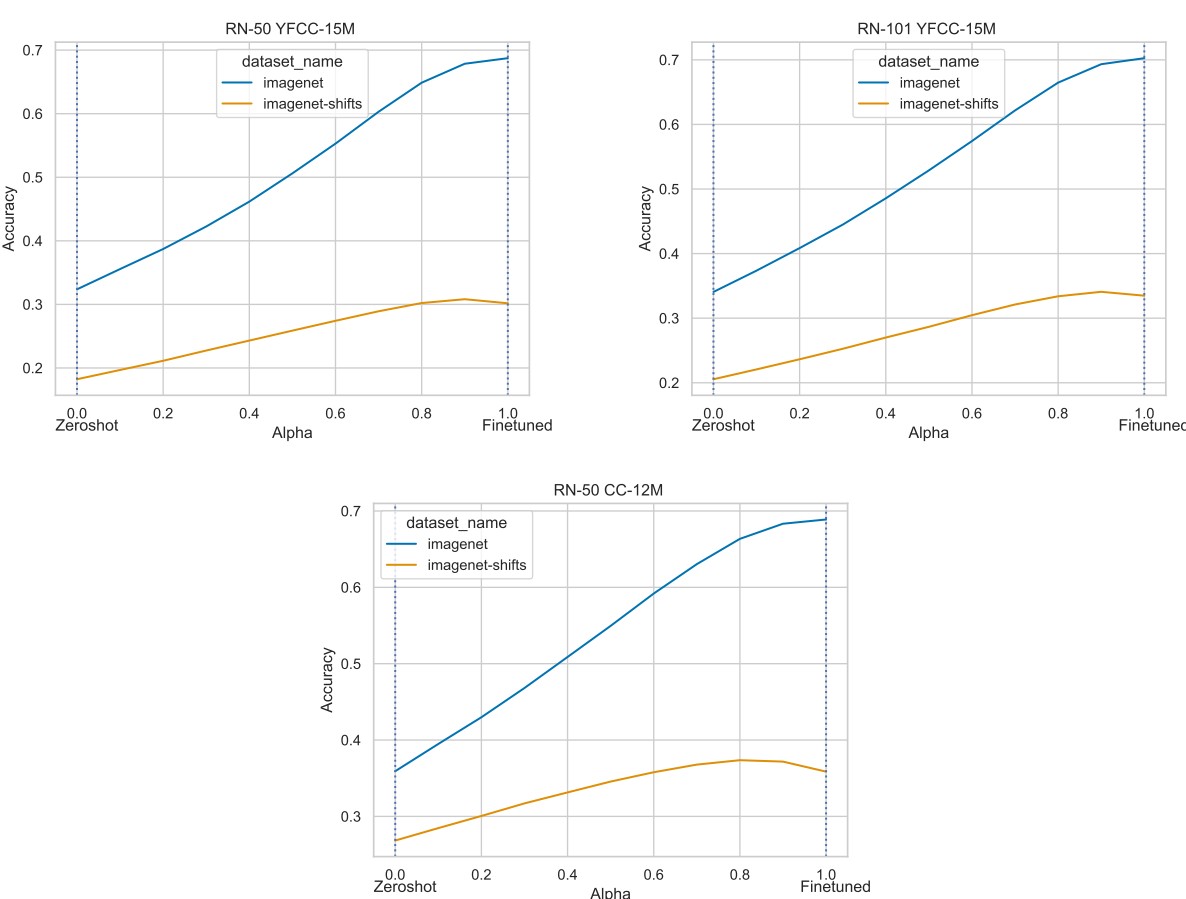

Figure 16: *Accuracies on ImageNet & shifts for Wise-FT models (2/2).*

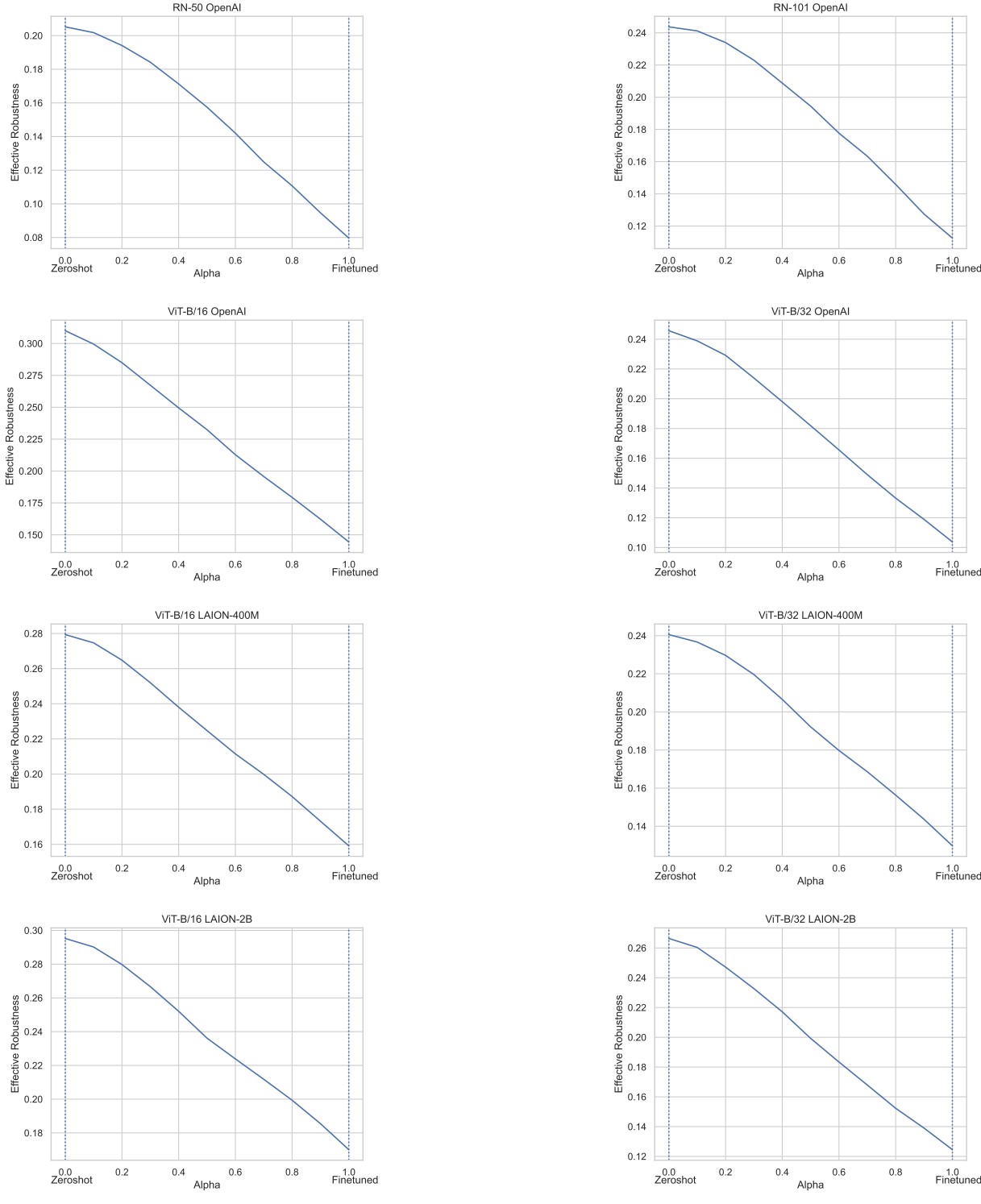

Figure 17: *ER for Wise-FT models (1/2).*

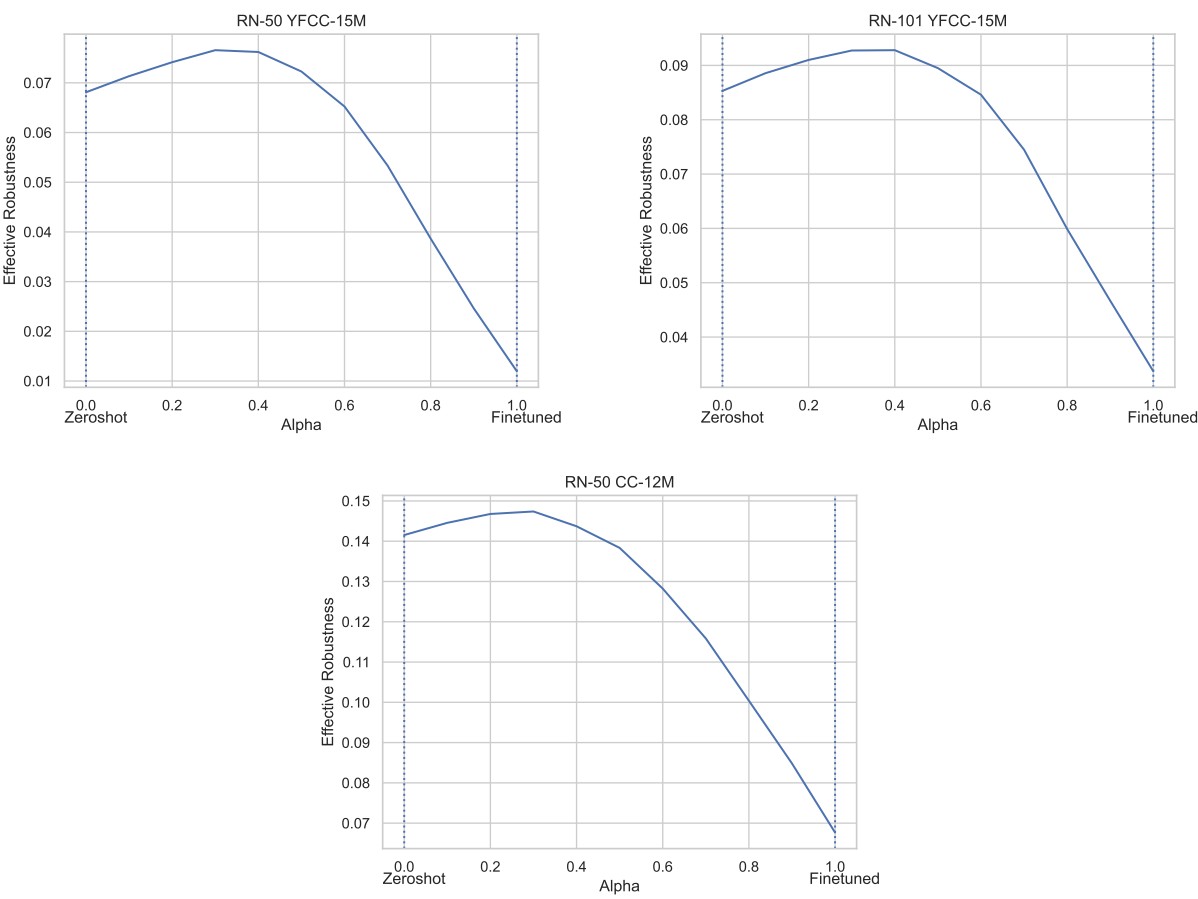

Figure 18: *ER for Wise-FT models (2/2).*

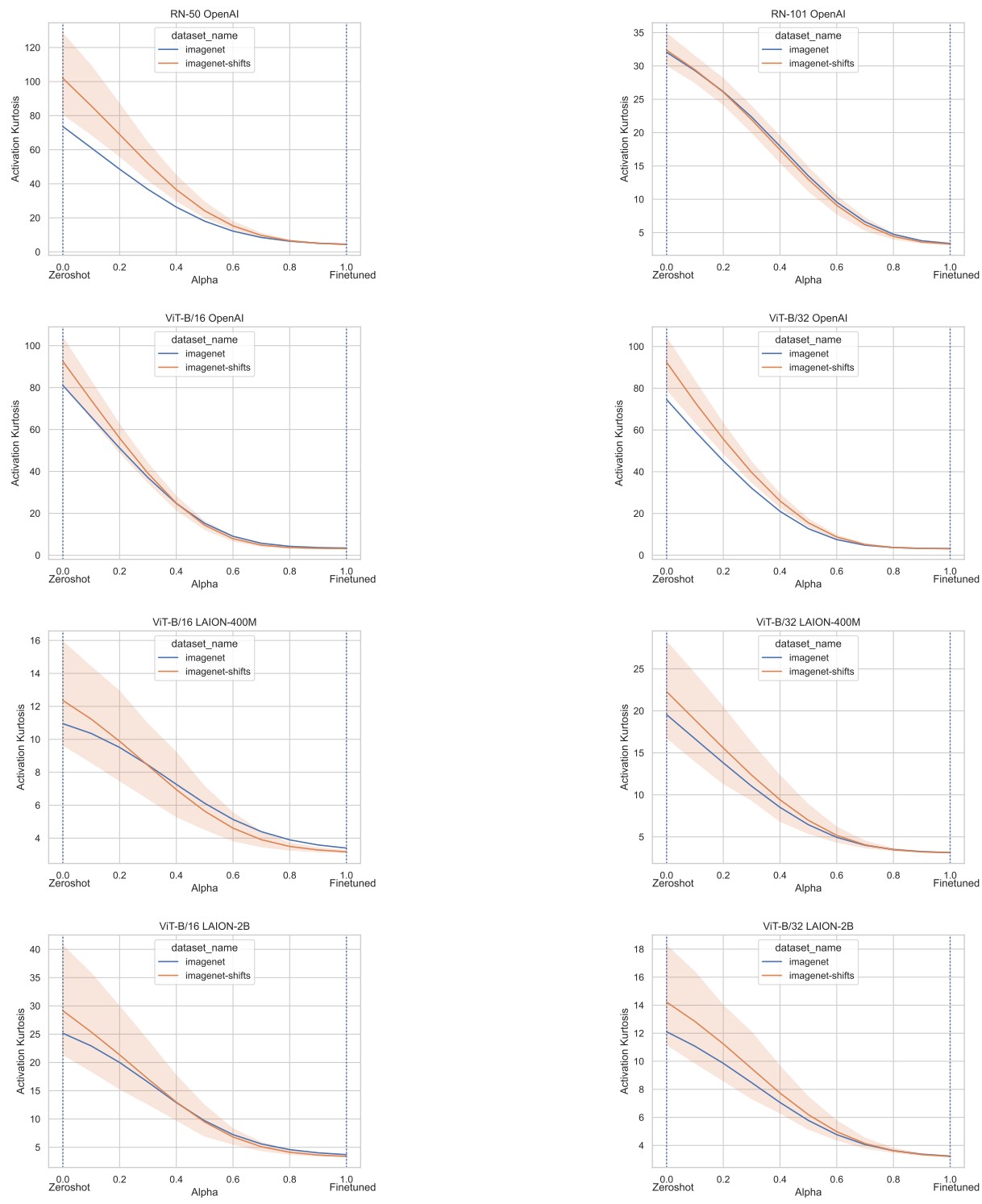

Figure 19: *Activation kurtosis for Wise-FT models (1/2). The activation kurtosis is computed for both ImageNet tests and ImageNet shifts.*

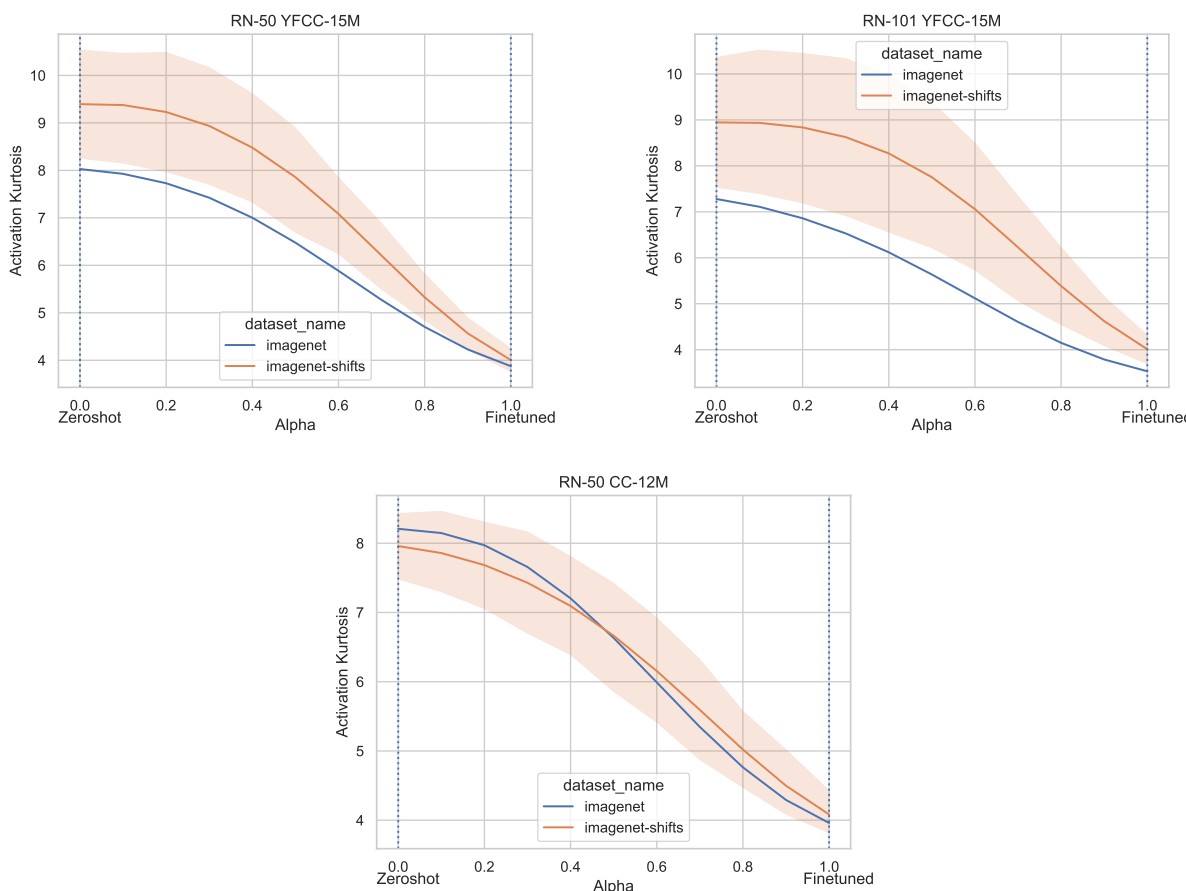

Figure 20: *Activation kurtosis for Wise-FT models (2/2). The activation kurtosis is computed for both ImageNet tests and ImageNet shifts.*

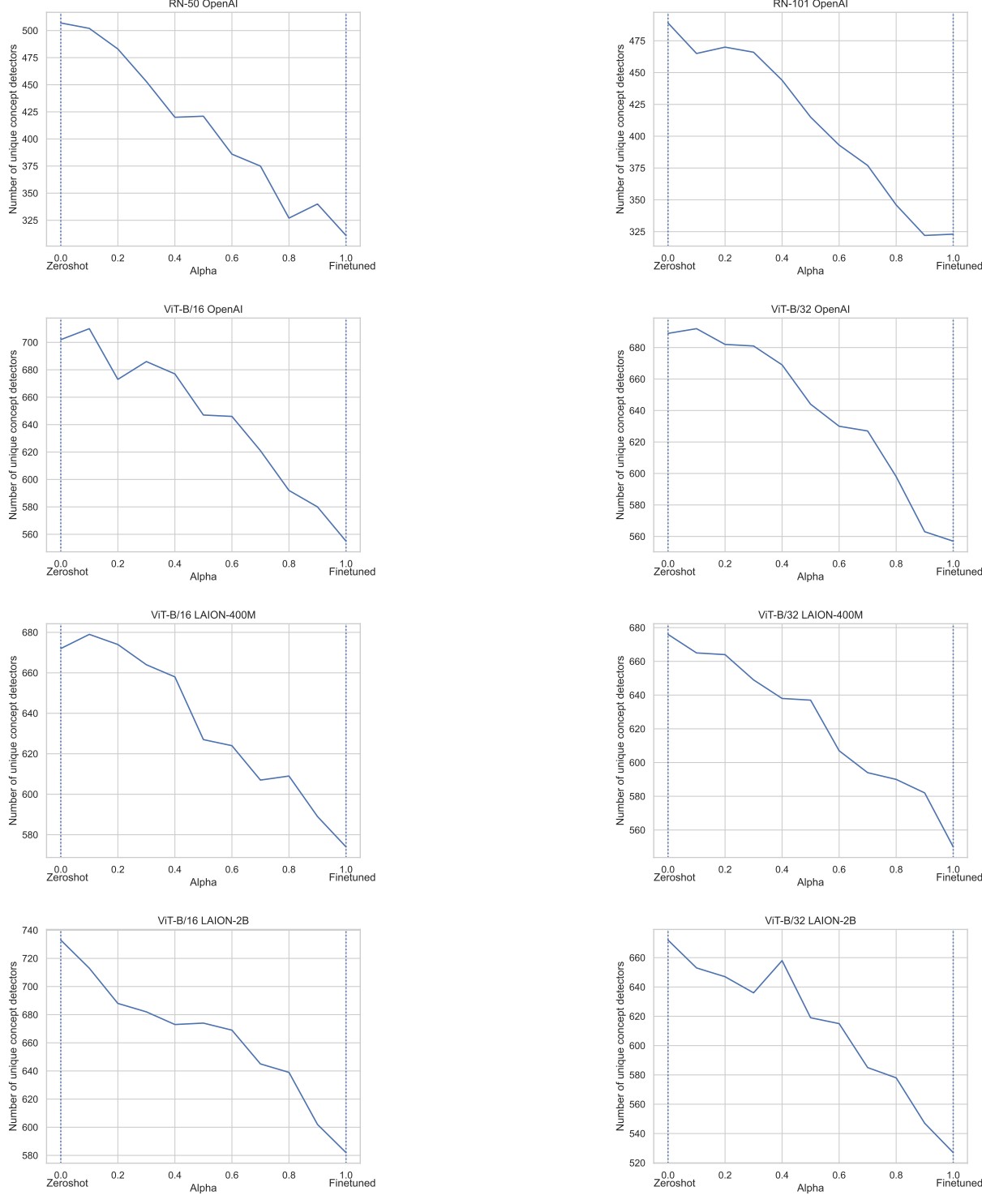

Figure 21: *Number of unique concepts encoded in Wise-FT models (1/2).*

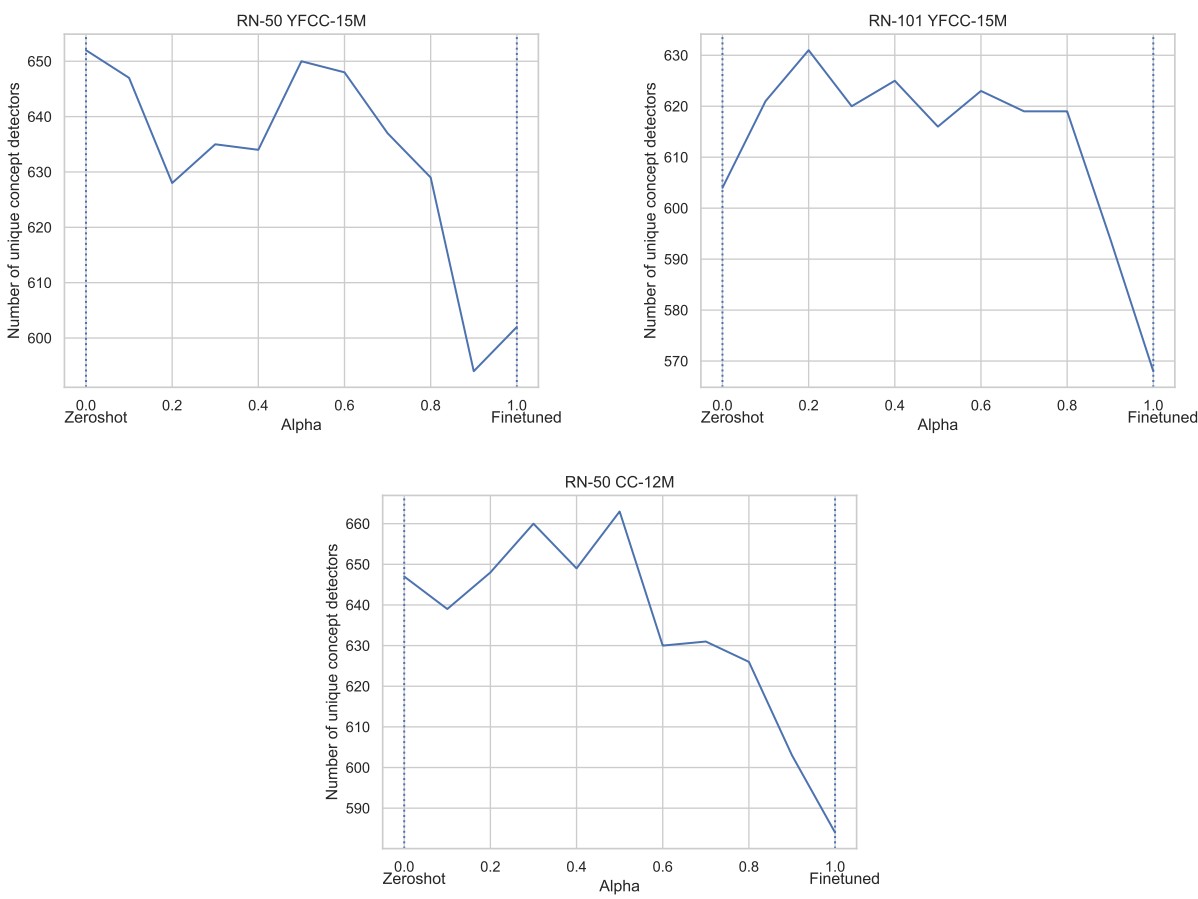

Figure 22: *Number of unique concepts encoded in Wise-FT models (2/2).*

## J   Further literature

**Defining CLIP ER.** The definition of ER crucially relies on the observation in multiple works that the model performance on natural shifts is linearly related to its performance in-distribution when both quantities are plotted with a logit scaling (Recht et al., 2018; 2019; Miller et al., 2020). We note though that there are known exceptions to this, e.g. considering out-of-distribution generalization on real-world datasets that substantially differ from the in-distribution dataset (Fang et al., 2023).

**Explaining CLIP ER.** A first intuitive explanation for the surprisingly high effective robustness of CLIP might be the fact that the learned embeddings are endowed with semantic grounding through pretraining with text data. This hypothesis was refuted by Devillers et al. (2021), who demonstrated that the embeddings in CLIP do not offer gains in unsupervised clustering, few-shot learning, transfer learning and adversarial robustness as compared to vision-only models. In a subsequent work, Fang et al. (2022) demonstrated that the high robustness of these models rather emerges from the high diversity of data present in their training set. This was achieved by showing that pretraining SimCLR models Chen et al. (2020) on larger datasets, such as the YFCC dataset by Radford et al. (2021), *without any language supervision* matches the effective robustness of CLIP. Shi et al. (2023) reinforced this data-centric explanation by showing that the performance on the pretraining set also correlates linearly with the out-of-distribution performance. To put the emphasis on the importance of data-quality for effective robustness, Nguyen et al. (2022) showed that increasing the pretraining set size does not necessarily improve the effective robustness of the resulting model. Rather, it suggests that it is preferable to filter data to keep salient examples, as was done, e.g., to assemble the LAION dataset (Schuhmann et al., 2022).

**Other indicators of ER.** By comparing pretrained models with models trained from scratch, Neyshabur et al. (2020) demonstrated that these models exhibit interesting differences, such as their reliance on high-level statistics of their input features and the fact that they tend to be separated by performance barriers in parameter space. Guillory et al. (2021) found observable model behaviours that are predictive of effective robustness. In particular, the difference of model's average confidence between the in and out-of-distribution correlates with out-of-distribution performance.

**Polysemanticity in foundation models.** Polysemantic neurons were coined by Olah et al. (2020) in the context vision model interpretability. These neurons get activated in the presence of several unrelated concepts. For instance, the InceptionV1 model has a neurons that fires when either cats or cars appear in the image. These neurons render the interpretation of the model substantially more complicated, as they prevent to attach unambiguous labels to all the neurons in a model. This will limit the insights gained by traditional interpretability techniques. For example, producing saliency maps for a polysemantic neuron could highlight many unrelated parts of an image, corresponding to the unrelated concepts this neuron is sensitive to. A qualitative analysis of the neurons in CLIP by Goh et al. (2021) showed that a CLIP ResNet has a substantial amount of polysemantic neurons. The emergence of polysemantic neurons is a complex phenomenon. It is not yet well-understood for models at scale. The latest works on the subject mostly focus on toy models, see e.g. the works of Elhage et al. (2022) and Scherlis et al. (2022). To the best of our knowledge, our work is the first to explicitly discusses the link that exists between polysemanticity and robustness to natural distribution shifts.

