# OpenReview forum: "Interpreting CLIP: Insights on the Robustness to ImageNet Distribution Shifts"
_TMLR — Accepted by TMLR_

### Review · Reviewer_ZEJm · 2024-05-15

**Summary Of Contributions:**

Robustness is a key property for machine learning models to be useful in practice. In short, it means that a desirable property in machine learning models is that when they see slightly different data compared to what they have been trained on, they should not lose too much of their performance on the original dataset.

ImageNet is a popular benchmark in image classification, with many natural shifts/slightly different variants of it. This gives us the opportunity to test if a model maintains its original performance on this slightly different variants, thereby measuring its robustness. From prior work, we have a large suite of models that work/do not work on this task. This paper asks the question: what is different between models that are robust and non-robust in this task?

The authors run a rigorous study of many different models and show the following:

1. **Robustness of clip models**: Pretrained clip models used in a zero-shot way are more robust than their fine-tuned counterparts.
2. **Outlier features**: features that have a significantly high magnitude compared to the average feature magnitude. Robust models have this more than non-robust models.
3. **Privileged direction**: Robust models have strong privileged direction compared to less-robust counterparts.
4. **Concept encoded**: Robust models encode more abstract concepts than non-robust ones, whose top-3 encoded concepts are more granular/fine-grained. Robust models also encode more concepts than their non-robust counterparts.

The paper uses prior methods to analyze these differences between robust and non-robust methods. While there is no causal relationship established (e.g., robustness implies existence of outlier features and vice versa), these observations can serve as easy to check conditions to estimate a model’s robustness.

**Audience:**

Yes

**Broader Impact Concerns:**

No concerns.

**Claims And Evidence:**

Yes

**Requested Changes:**

Please answer the weaknesses/questions mentioned above.

**Strengths And Weaknesses:**

## **Strengths**

1. The paper is nicely structured and well-written: it studies multiple different concepts, but they are well-organized with sections and in particular take-away boxes, which would help readers/practitioners.
2. The experiment suite of the paper is extensive and thorough. The authors test 5 backbone architectures (ResNet50, ResNet101, ViT-B/16, ViT-B/32, ViT-L/14) and many different variations (clip pretraining, fine-tuning, pretrained with ImageNet captions). This provides a very thorough picture of different attributes of robustness of all these models. To the best of my knowledge, this is the most comprehensive study of all these models. While lacking novelty, the paper can serve as a nice survey of all these pre-trained models and their robustness attributes to practitioners working on similar domains.
3. The set of observations, especially that language supervision leads to outlier features and more robust models encode more abstract concepts in comparison to fine-tuned/less robust models, is very interesting. There is potential for future research based on these observations on how to maintain robustness while fine-tuning a pre-trained model for a specific task, and how to get signal for robustness without having a test set, by measuring other attributes such as outlier features.

In short, **I am happy with accepting the paper with minimal changes**, provided all my questions are answered.


## **Weaknesses**

**(Section 3)**

First, this way of measuring effective robustness (by comparing the performance of the model to the predicted performance based on accuracy-on-the-line) is a bit alien to me. The reason we call in-distribution performance “in-distribution”, is because this is the distribution that the model has been trained on. If the only way to break out of the linear trend is to use more diverse data, then we should consider performance on the more diverse data as “in-distribution performance”. In other words, if a model is trained on something other than ImageNet (possibly a superset, but even if it is not, the point stands), then performance on ImageNet cannot be considered as “ID performance.” For zero shot models (no fine-tuning), if our training distribution is completely unknown, we can as easily call the natural shift data as “ID” and the original ImageNet data as OOD. What does it say about the effective robustness metric? How do the numbers look in this case? **However, I note that prior work has also done it the same way, and the authors would point to [1], and so this is not a weakness of the current work.**

What is a weakness of the current work is mentioning that zero-shot clip models are more robust than their fine-tuned counterparts. This has been studied, both conceptually and empirically, by [2]. They also compare other methods such as linear-probing and LP-FT. Prior work such as [3, 4] also showed effectiveness of tuning particular layers instead of the entire network. I suggest the following experiments:

1. comparison to linear-probing and LP-FT
2. Add a linear head on top of the visual encoder with C number of classes, and fine-tune only that with the ImageNet images



## **Questions**

**(Contributions)**

> To the best of our knowledge, this is the first time that outlier features are observed in non-language and non-transformer models.

Could the authors cite papers that have observed this for language or transformer models? This citation is added later, in section 4, but should be added to the introduction as well.

**(Nitpick, activation vectors)**

Does the image encoder $f_v$ output the activation vector directly? Or does it output an embedding $e$, that passes through an activation function $\sigma$, to get the activation vector? This does not make much difference to the analysis, I am just curious about the notations.

# References

References

[1] Measuring Robustness to Natural Distribution Shifts in Image Classification, https://arxiv.org/abs/2007.00644

[2] Fine-Tuning can Distort Pretrained Features and Underperform Out-of-Distribution, https://arxiv.org/abs/2202.10054

[3] Surgical Fine-Tuning Improves Adaptation to Distribution Shifts, https://arxiv.org/abs/2210.11466

[4] Less is More: Selective Layer Finetuning with SubTuning, https://arxiv.org/abs/2302.06354

---

### Review · Reviewer_SLo8 · 2024-06-26

**Summary Of Contributions:**

This paper studies what differentiates robust models from non-robust ones. The authors analyze a range of CLIP models with varying backbones, pretraining datasets, and robustness levels using different interpretability tools. They identify three key findings:
1. Outlier features in robust CLIP models: Robust CLIP models exhibit outlier features with significantly higher activations compared to other features.
2. Outlier features as robustness signatures: The presence of outlier features consistently distinguishes robust models from non-robust ones. This suggests that outlier features can serve as an indicator of robustness.
3. Concept richness is not solely linked to robustness: While robust CLIP models encode a high number of unique concepts, this characteristic is also observed in non-robust models trained on ImageNet-Captions. This implies that language supervision plays a significant role in enriching visual representations with human concepts.

**Audience:**

Yes

**Claims And Evidence:**

Yes

**Requested Changes:**

Adding an analysis of what these outlier features represent.

**Strengths And Weaknesses:**

Strengths:
1. Novel findings: The paper presents novel insights into the relationship between feature characteristics and robustness in CLIP models.
2. Comprehensive analysis: This study provides a comprehensive analysis of the robustness of CLIP models by utilizing various models.
3. The paper is well-written and easy to follow. The analysis is presented logically.

Weaknesses:
1. Limited explanation of outlier features: While the paper identifies outlier features as a key finding, it doesn't delve deep enough into their potential causes or implications. Further investigation into what these outlier features represent and why they emerge in robust models would significantly strengthen the paper.
2. Limited dataset: The study primarily focuses on ImageNet distribution shifts. Exploring the generalizability of the findings to other datasets would be better.

---

> ### Author Response · Authors · 2024-08-01
>
> We would like to thank the reviewer for their time and useful feedback! We appreciate that the reviewer has identified several strengths of our paper, and would like to take the opportunity to briefly comment on the weaknesses mentioned.
>
> **Limited dataset**: We agree with the reviewer that exploring the generalizability of our findings to other datasets beyond ImageNet distribution shifts would be interesting - in fact we mention this in the last paragraph of our paper as an interesting direction for future work.
> For the scope of this work, we only make claims about ImageNet distribution shift robustness, for which we provide evidence. Thus, we believe this analysis to be beyond the scope of the current work, and would leave it for future work.
>
> **Explanation of outlier features**: Regarding the further investigation into outlier features, we would like to point out that we do give some insights into potential causes and implications of outlier features at the end of Section 4 (Paragraphs ‘Emergence of outlier features.’ and ‘Relationship of outlier features to pruning.’). We agree with the reviewer that a deeper investigation into this would be very interesting, and hope to inspire future work on this with the findings of our paper. However, we are happy to discuss this point further and potentially perform additional experiments should the reviewer find them necessary for validating the claims of our paper.

---

### Review · Reviewer_aZBL · 2024-07-21

**Summary Of Contributions:**

This paper investigated what makes robust language-image models different from non-robust ones, particularly in handling ImageNet distribution shifts. The author probed the representation spaces of 16 robust CLIP vision encoders with various backbones and pretraining sets, comparing them to less robust models with identical backbones but different (pre)training sets or objectives. This paper provided empirical observations and studied the relationships between robustness and various concepts, such as outlier features, privileged directions, and the number of unique concepts.

**Audience:**

Yes

**Claims And Evidence:**

Yes

**Requested Changes:**

Please further clarify how the takeaway messages are supported by empirical evidence.

**Strengths And Weaknesses:**

## Strengths

- The empirical observations provided in this paper are insightful, including:
  - All robust models have **outlier features**;
  - **Privileged directions** are crucial to model predictions, but they can be also found in non-robust models;
  - Robust models have a high number of **unique concepts**.
- This paper is well written and contextualized.

## Weaknesses

- Some statements were expressed too vaguely. My understanding is that the author empirically showed that "for all models, being robust $\to$ having outlier features" and "for all models, not being robust $\to$ not having outlier features" hold. However, "there exists a model, not being robust $\to$ having privileged directions" is also true. It would be nice if the author could clarify what "being a signature of robustness" precisely means.
- Some claims were not supported by sufficient evidence. For example, the evidence of the takeaway message 3 seems too weak. I'm aware that training large models is computationally costly, but the presented evidence seems insufficient to draw a conclusion. Anyway, the difference between generic/abstract concepts and concrete concepts is subtle and too subjective. Further, I may have missed something, but the connection between the experimental results and the claims in the takeaway message 4 is unclear to me. How is the statement "high number of concepts stems from language supervision" supported and verified?
- The post hoc analysis and insights are valuable, but this paper would be more impactful if the author could discuss more how we can improve the robustness and generalization ability of language-image models based on the given insights.

---

> ### Author Response · Authors · 2024-08-01
> **Clarifications and adjustments**
>
> We thank the reviewer for their time and useful feedback.
> We are pleased to hear that the reviewer finds the paper insightful, well-written, and of interest to TMLR’s audience.
> More importantly, we want to take the opportunity to clarify and adjust some of the claims in our paper based on the reviewer’s feedback.
>
> **Re W1 (Ambiguity of term ‘signature’):** We agree with the reviewer that it would be beneficial for the clarity of the manuscript to better define what  ‘being a signature’ means. We thank the reviewer for pointing out this potential source of ambiguity, and would like to clarify this aspect both here and in the manuscript.
>
> To clarify, we use the term signature as follows: $A$ being a signature of $B$ means that $A \implies B$, and $\neg B \implies \neg A$ equivalently.
> For the statements in our paper, this corresponds to the following: The presence of outlier features implies robustness of the model, or equivalently a non-robust model will not have outlier features. Since we do not find outlier features in any of the non-robust models investigated, we claim that based on our experiments outlier features appear to be such a signature of robustness (see Take-away 2).
> For privileged directions however, we find some models (those pretrained on ImageNet-Captions) that do have privileged directions but are not robust. Therefore, we deduce that they are not a signature of robustness (see Take-away 2).
> We have updated the manuscript at crucial places (abstract, contributions, take-aways, conclusion) to clarify this, including adding a footnote on page 1 that explains our usage of the term signature in this paper.
>
> **Re W2.1 (Insufficient evidence for Take-away 3):** Regarding Take-away 3, we agree with the reviewer that for any qualitative observation, there can always be a subjective debate about if there is enough evidence since it is not easily measurable (as opposed to quantitative observations). We made this specific observation about the nature of the encoded concepts when analyzing the results in Table 6 in Appendix E.3 (which we refer to and summarize for the OpenAI models in the main part of the paper) and think it is a curiosity that we want to share with the reader.
> We would have two suggestions on how to adjust its presentation to make it more in line with the evidence:
> - *Option A*: Keep as take-away box, but rephrase as ‘We qualitatively observe that privileged directions of robust zero-shot CLIP models tend to encode rather generic texture information, while fine-tuning tends to replace these generic concepts in privileged directions by less abstract and more concrete concepts.’
> - *Option B*: Remove take-away box, and replace with the following paragraph in plain text: ‘We can find similar patterns for the remaining models in Table 6 (Appendix E.3). In summary, we observe that qualitatively, privileged directions of robust zero-shot CLIP models tend to encode rather generic texture information, while fine-tuning tends to replace these generic concepts in privileged directions by less abstract and more concrete concepts.’
>
> Would the reviewer agree with the presentation of our observation in one of those two forms?
>
>
> **Re W2.2 (Evidence for Take-away 4):** The evidence and analysis that leads us to this statement is described at the end of page 9 / beginning of page 10 (after Eq (4)) in the original submission, and we summarize it here for clarity: We first observe that all zero-shot models (including the non-robust ImageNet-Captions CLIP models) encode more concepts than the supervised models for each architecture. Then, we observe that among the ImageNet-Captions CLIP models, the amount of concepts varies depending on the type of language supervision used (title, title+description, or title+description+tag). Both of these observations support the idea that language supervision influences the number of concepts encoded in the features, rather than the number of concepts being related to the robustness of the model.

---

> > ### Comment · Reviewer_aZBL · 2024-08-04
> >
> > Thank you for the clarification. I believe incorporating these revisions can make this paper clearer.
> >
> > I don't have any remaining concerns or questions.

---

> > > ### Author Response · Authors · 2024-08-05
> > >
> > > We would like to thank the reviewer again for the useful feedback that has led to the clarifications above. We agree with the reviewer that incorporating them into the paper can make the paper clearer and have therefore done so in the updated manuscript of this submission (downloading the pdf now should download that updated version).
> > >
> > > If there are no remaining concerns or questions, could we kindly ask the reviewer to change their answer to 'Claims And Evidence' in the original review from 'No' to 'Yes'?

---

### Decision · Action_Editor_LSzo · 2024-10-01

**Recommendation:** Accept with minor revision

**Comment:**

As I mentioned above, some of the conclusions in the paper are not presented very clearly, and may be lacking evidence to support them in the current version. I believe the paper should be accepted, but I request that the authors update the wording in the paper, especially the take-away boxes to clarify that (1) no causal conclusions can be made and (2) to incorporate the suggestions of Reviewer aZBL for takeaway box 4.

**Audience:**

Robustness and interpretability are both central topics in modern deep learning, and the paper is very relevant to the TMLR community.

**Claims And Evidence:**

The paper studies the feature representations learned by CLIP models with various architectures and pretraining datasets in the context of robustness. The authors use techniques inspired by language model interpretability research such as identifying outlier features, studying polysemanticity etc. The authors show several takeaways:
- Outlier features, i.e. features with activations substantially exceeding the average, are present in all robust models and not present in all non-robust models considered in the paper.
- The authors provide a nice study of the human concepts encoded by various CLIP models pre- and post-finetuning on ImageNet.
- The authors also study polysemanticity and show that robust models encode more unique concepts, but argue that this is due to the language supervision.

Overall, the paper provides an interesting study, and most claims are supported by envidence. However, some of the results seem inconclusive, and the evidence is insufficient to make strong claims. In particular,
- It seems possible that the presence of outlier features is the **effect** of training on the diverse dataset, and not the **cause** of robustness. While the authors may not be claiming a causal statement, it seems to be suggested by the "signature of robustness" terminology.
- Reviewer aZBL also raises a point about the conclusion of Section 4 that states that the number of concepts is related to language supervision and not the model robustness. The authors agreed with this criticism and suggested a reasonable update to the takeaways.